# c-Myc inactivation of p53 through the pan-cancer lncRNA MILIP drives cancer pathogenesis

Yu Chen Feng [1,8], Xiao Ying Liu [2,8], Liu Teng[2], Qiang Ji[2], Yongyan Wu [3], Jin Ming Li[2], Wei Gao [3], Yuan Yuan Zhang [1], Ting La[1], Hessam Tabatabaee [1], Xu Guang Yan [1], M. Fairuz B. Jamaluddin [1], Didi Zhang[4], Su Tang Guo[5], Rodney J. Scott [1], Tao Liu[6], Rick F. Thorne [1,2], Xu Dong Zhang [1,2✉] & Lei Jin [2,7✉]

The functions of the proto-oncoprotein c-Myc and the tumor suppressor p53 in controlling cell survival and proliferation are inextricably linked as "Yin and Yang" partners in normal cells to maintain tissue homeostasis: c-Myc induces the expression of ARF tumor suppressor (p14[ARF] in human and p19[ARF] in mouse) that binds to and inhibits mouse double minute 2 homolog (MDM2) leading to p53 activation, whereas p53 suppresses c-Myc through a combination of mechanisms involving transcriptional inactivation and microRNA-mediated repression. Nonetheless, the regulatory interactions between c-Myc and p53 are not retained by cancer cells as is evident from the often-imbalanced expression of c-Myc over wildtype p53. Although p53 repression in cancer cells is frequently associated with the loss of ARF, we disclose here an alternate mechanism whereby c-Myc inactivates p53 through the actions of the c-Myc-Inducible Long noncoding RNA Inactivating P53 (MILIP). MILIP functions to promote p53 polyubiquitination and turnover by reducing p53 SUMOylation through suppressing tripartite-motif family-like 2 (TRIML2). MILIP upregulation is observed amongst diverse cancer types and is shown to support cell survival, division and tumourigenicity. Thus our results uncover an inhibitory axis targeting p53 through a pan-cancer expressed RNA accomplice that links c-Myc to suppression of p53.

[1] School of Biomedical Sciences and Pharmacy, The University of Newcastle, Newcastle 2308 NSW, Australia. [2] Translational Research Institute, Henan Provincial People's Hospital and People's Hospital of Zhengzhou University, Academy of Medical Science, Zhengzhou University, Zhengzhou, 450053 Henan, China. [3] Department of Otolaryngology, Shanxi Key Laboratory of Otorhinolaryngology Head and Neck Cancer, the first affiliated hospital, Shanxi Medical University, Taiyuan 030001 Shanxi, China. [4] Department of Orthopaedics, John Hunter Hospital, Hunter New England Health, Newcastle 2305 NSW, Australia. [5] Department of Molecular Biology, Shanxi Cancer Hospital and Institute, Taiyuan, 030013 Shanxi, China. [6] Children's Cancer Institute Australia for Medical Research, University of New South Wales, Sydney 2750 NSW, Australia. [7] School of Medicine and Public Health, The University of Newcastle, Newcastle 2308 NSW, Australia. [8] These authors contributed equally: Yu Chen Feng, Xiao Ying Liu. ✉email: Xu.Zhang@newcastle.edu.au; Lei.Jin@newcastle.edu.au

$M$ YC is arguably the best characterized proto-oncogene that is aberrantly activated in ~40% of human cancers by chromosomal translocation, gene amplification, and by upstream oncogenic signals[1–3]. Its product, c-Myc, is a transcription factor that regulates thousands of genes involved in numerous cellular functions including cell survival and proliferation[4,5]. On the other hand, *TP53*, encoding the best-known tumor suppressor p53, plays a pivotal role in response to cellular stress, in particular genotoxic stress, leading to either DNA repair or elimination of cells with severe DNA damage[6,7]. c-Myc and p53 interact in a negative feedback manner to maintain cellular homeostasis under physiological conditions[8,9]. While c-Myc induces the expression of ARF tumor suppressor (p14^ARF in human and p19 ^ARF in mouse) that binds to and inhibits the E3 ubiquitin–protein ligase mouse double minute 2 homolog (MDM2) leading to p53 activation[10,11], p53 transcriptionally inactivates c-Myc and also represses c-Myc through microRNA-mediated mechanisms[12,13].

The frequently imbalanced expression of c-Myc over p53 signifies that the regulatory interactions between c-Myc and p53 are paralyzed in cancer cells[14]. Although inactivation of *TP53* through mutation occurs in ~50% of human cancers, the expression of p53 in *TP53* wild-type cancers remains low even in the presence of high levels of c-Myc[15,16]. This is closely associated with the loss of p14^ARF through mutation and hypermethylation of its coding gene *CDKN2A*[17,18], but whether other mechanisms are involved is not known. Notably, apart from ubiquitination, p53 can be modified post-translationally through other mechanisms, including phosphorylation, acetylation, and SUMOylation, which contribute to regulation of its expression and activity[19,20].

The role of long noncoding RNAs (lncRNAs) in cancer development and progression is increasingly appreciated[21,22]. In particular, the list of lncRNAs involved in the c-Myc and p53 pathways in cancer cells is rapidly expanding[23–25]. For example, the c-Myc-responsive lncRNA isocitrate dehydrogenase (IDH) 1 antisense RNA1 (IDH1-AS1) supresses cancer cell proliferation through a metabolic mechanism[26], whereas p53 regulates the expression of the lncRNA GUARDIN that is essential for genomic stability and thus promotes cancer cell survival[27]. Of particular interest, c-Myc transcriptionally activates the expression of the lncRNA SENEBLOC that functions as a scaffold to facilitate the binding of MDM2 with p53 leading to downregulation of p53 and prevention of cell senescence[28].

Here, we demonstrate that c-Myc can alternatively inactivate p53 through the lncRNA MILIP that restrains p53 SUMOylation through suppressing the SUMO E3 ligase tripartite-motif family-like 2 (TRIML2) and thus facilitates p53 polyubiquitination and turnover. Moreover, we show that MILIP is commonly upregulated across diverse cancer types and is critical for cancer cell survival, division and tumourigenicity, with practical implications of targeting MILIP as a therapeutic avenue for cancer treatment.

## Results

**c-Myc drives MILIP expression that is upregulated in diverse cancer types.** Since aberrant c-Myc activation drives key gene networks contributing to the pathogenesis of many human cancers[5], and an increasing number of lncRNAs have been found to be involved in c-Myc-mediated signaling[23], we sought to identify lncRNA effectors downstream of c-Myc that may play roles in cancer development and progression in the pan-cancer context. Through interrogating the lncRNA expression data in the Cancer Genome Atlas (TCGA)[29], we identified a panel of lncRNAs that were upregulated in at least 18 of 20 cancer types in relation to corresponding normal tissues (Supplementary Fig. 1a). Among

the five most prominently upregulated lncRNAs were two previously reported c-Myc-responsive lncRNAs, CONCR (DDX11-AS1), and GAS5[30,31]. A third lncRNA that we now call MILIP [also known as v-maf avian musculoaponeurotic fibrosarcoma oncogene homolog G (MAFG) antisense RNA1 (MAFG-AS1)] also contains a consensus c-Myc-binding region (c-Myc-BR) in its gene promoter (Supplementary Figs. 1a, b and 2a). We, therefore, set to investigate whether MILIP is indeed regulated by c-Myc and whether it is involved in cancer pathogenesis.

Analysis of chromatin immunoprecipitation sequencing (ChIP-seq) data from the Encyclopedia of DNA Elements (ENCODE) Consortium revealed c-Myc and Max, which is necessary for c-Myc-mediated transcriptional activation[32], binding peaks at the c-Myc-BR of the *MILIP* promoter (Supplementary Fig. 2a). In accordance, c-Myc co-precipitated the *MILIP* promoter and siRNA silencing of c-Myc reduced the expression of MILIP along with the lncRNA OVAAL known to be regulated by c-Myc in human A549 lung adenocarcinoma, MCF-7 breast cancer, and HCT116 colon cancer cells (Fig. 1a, b, Supplementary Fig. 2b, c)[33]. Consistently, induced inhibition of c-Myc in P493-6 human B-cell lymphoma cells resulted in a decrease in MILIP expression (Supplementary Fig. 2d)[34]. On the other hand, c-Myc overexpression upregulated MILIP, which was however diminished by knockdown of Max (Supplementary Fig. 2e, f)[32]. Moreover, c-Myc silencing diminished transcriptional activity of the luciferase reporters containing the intact c-Myc-BR of *MILIP* promoter, whereas c-Myc overexpression selectively enhanced reporter activity (Fig. 1c, Supplementary Fig. 2g, h). Further substantiating the role of c-Myc in transcriptional activation of MILIP, CRISPR/Cas9-mediated deletion of the c-Myc-BR at the endogenous *MILIP* gene promoter diminished the expression of MILIP, which could not be rescued by c-Myc overexpression (Supplementary Fig. 2i, j). Thus, MILIP expression is driven by c-Myc in a range of cancer cell types. In support, MILIP expression levels correlated with *MYC* gene expression in diverse human cancer types (Supplementary Fig. 2k). Notably, siRNA knockdown of c-Myc did not cause any significant changes in the MILIP levels in MCF10A and HME-1 untransformed human normal mammary epithelial cells (Supplementary Fig. 2l). However, overexpression of c-Myc upregulated MILIP expression in these cell types (Supplementary Fig. 2m), implicating the involvement of mechanisms other than c-Myc in regulating the expression of MILIP in normal cells.

Instructively, MILIP expression was increased in cancer versus normal cell lines as shown in comparisons between human MDA-MB-231 breast cancer and U2OS osteosarcoma cells in addition to A549, HCT116, and MCF-7 cells compared with MCF10A and HME-1 cell lines (Supplementary Fig. 3a). Indeed, absolute quantitation assays demonstrated that there were ~79–152 MILIP molecules per cancer cell compared with ~23–30 MILIP molecules per mammary epithelial cell (Supplementary Fig. 3b). Similarly, MILIP was upregulated in cohorts of non-small cell lung carcinoma and colon cancer tissues compared with paired adjacent normal epithelial tissues (Fig. 1d, e, Supplementary Fig. 3c–e, Supplementary Tables 1 and 2).

The gene encoding MILIP is located head to head with the *MAFG* gene on chromosome 17 separated by a short distance of 117 bp (Supplementary Fig. 4a). Nevertheless, neither knockdown nor overexpression of MILIP impinged upon MAFG expression (Supplementary Fig. 4b, c) and similarly, knockdown or overexpression of MAFG also failed to affect MILIP expression levels (Supplementary Fig. 4d, e). Collectively, this indicates there is no regulatory interplay between MILIP and its neighboring gene *MAFG*[35]. Moreover, in contrast to the regulation of MILIP by c-Myc (Fig. 1b, Supplementary Fig. 2c–f), knockdown or overexpression of c-Myc in A549 and MCF-7 cells did not alter the expression of MAFG mRNA (Supplementary Fig. 4f, g),

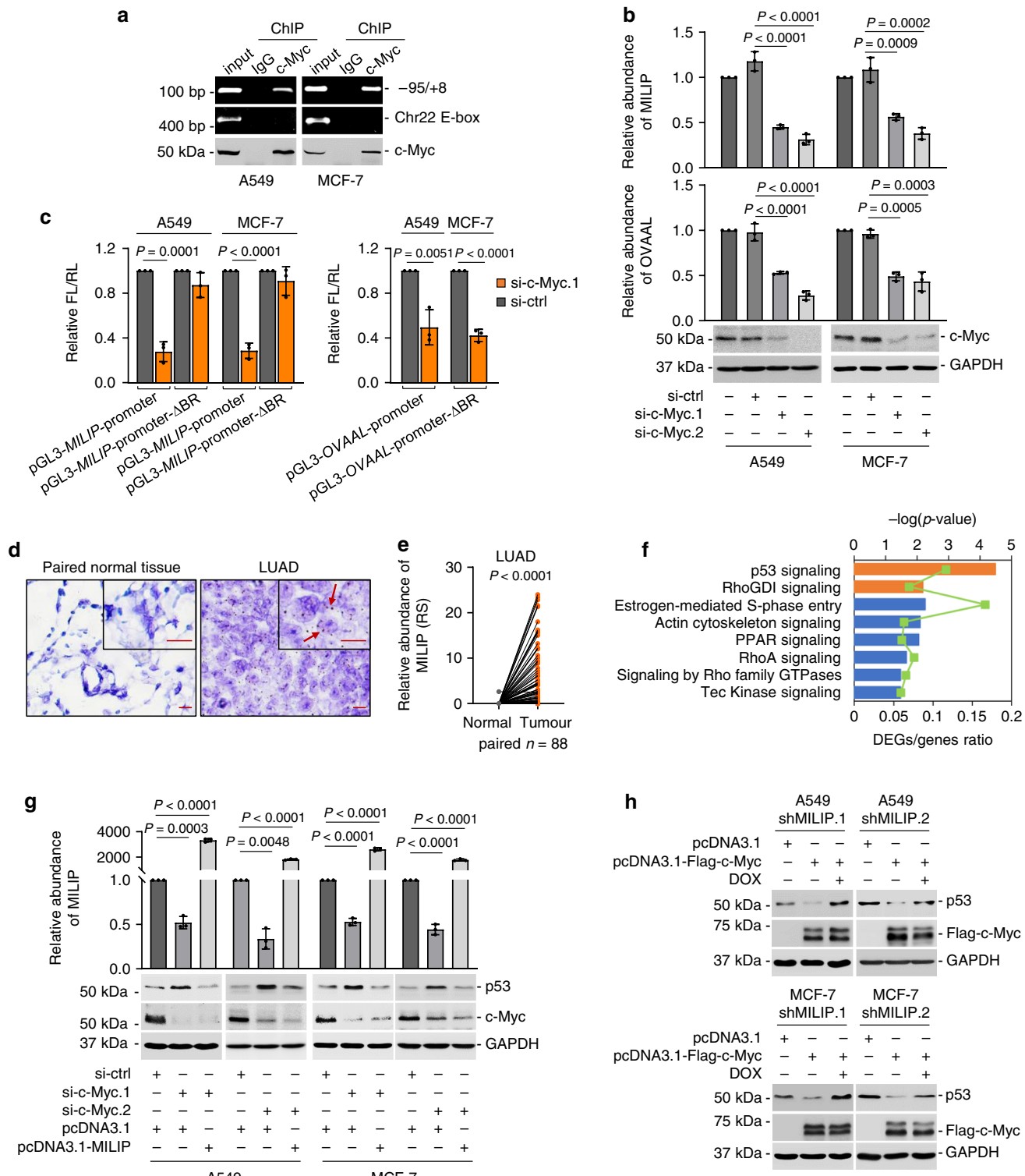

suggesting that despite their close proximity, c-Myc selectively transactivates MILIP but not MAFG.

MILIP has two annotated isoforms of 1895 and 969 bp, respectively (Vega Genome Browser) (Supplementary Fig. 4h). PCR analyses with exon-specific primers readily detected expression of the longer isoform (MILIP-001) in A549, MCF7, and HCT116 cells whereas the shorter isoform (MILIP-002) could not be measured (Supplementary Fig. 4i). The long isoform consists of a small exon (E1) and a large exon (E2), with

minimum free energy modeling predicting a symmetrical structure with each exon primarily contributing to one pole of the molecule (Supplementary Fig. 4j)[36].

**c-Myc represses p53 through MILIP.** RNA-sequencing analysis of A549 cells following siRNA knockdown of MILIP revealed that p53 signaling was the most enriched gene pathway (Fig. 1f, Supplementary Fig. 5a). In support, induced knockdown of MILIP increased p53 expression and its transcriptional activity in

**Fig. 1 c-Myc activates the pan-cancer lncRNA MILIP to repress p53. a** c-Myc bound to *MILIP* promoter in A549 and MCF-7 cells. An E-box motif, not associated with MYC target genes on Chr22, was used as a negative control. Data shown represent three independent experiments. ChIP, chromatin immunoprecipitation. **b** c-Myc silencing downregulated MILIP expression in A549 and MCF-7 cells. c-Myc responsive lncRNA OVAAL was used as a positive control. Data are mean ± s.d.; n = 3 independent experiments, one-way ANOVA followed by Tukey's multiple comparisons test. **c** c-Myc silencing reduced the activity of reporters with intact c-Myc binding region (BR) of *MILIP* promoter but not that with the c-Myc-BR deleted (ΔBR) in A549 and MCF-7 cells. Data are mean ± s.d.; n = 3 independent experiments, two-tailed Student's *t* test. **d** Representative microscopic photographs of in situ hybridization (ISH) analysis of MILIP expression in formalin-fixed paraffin-embedded (FFPE) lung adenocarcinoma (LUAD; n = 88 biologically independent samples) compared with paired adjacent normal tissues. Scale bar, 5 μm. **e** Quantitation of MILIP expression as detected in (**d**) in FFPE LUAD in comparison with paired adjacent normal tissues. RS, reactive score. Two-tailed Student's *t* test. **f** Ingenuity Pathway Analysis (IPA) of RNA-seq data showing that p53 signaling was the most enriched pathway in A549 cells transfected with a MILIP siRNA (si-MILIP.2) relative to those introduced with the control siRNA. Orange bars represent pathways that were activated, and blue bars, inactivated. DEGs differentially expressed genes. **g** c-Myc knockdown upregulated p53 expression, which was diminished by MILIP overexpression in A549 and MCF-7 cells. Data are representatives or mean ± s.d.; n = 3 independent experiments, One-way ANOVA followed by Tukey's multiple comparisons test. **h** c-Myc overexpression downregulated p53 expression, which was abolished by knockdown of MILIP in A549 and MCF-7 cells. Data shown represent three independent experiments. DOX, doxycycline, 200 ng/mL. Source data of Fig. 1a–c, e–h are provided as a Source Data file.

A549 and MCF-7 cells carrying an inducible MILIP shRNA system responsive to doxycycline (DOX) (Supplementary Fig. 5b–d), whereas overexpression of MILIP reduced p53 expression and its transcriptional activity (Supplementary Fig. 5e–g). The increase in a subset of p53 downstream targets caused by MILIP knockdown was confirmed in A549 cells (Supplementary Fig. 5h–j). Conversely, MILIP overexpression caused, albeit moderately, reduction in the expression of these p53 downstream targets (Supplementary Fig. 5k). CRISPR-interference (CRISPRi) silencing of MILIP upregulated p53 expression (Supplementary Fig. 5l), further substantiating the role of MILIP in regulating p53 expression. In contrast, p53 did not have a role in regulating MILIP expression (Supplementary Fig. 5m), as siRNA knockdown of p53 did not affect MILIP expression levels. These results, along with the finding that c-Myc drives MILIP expression, suggest that in contrast to upregulating p53 in normal cells[11], c-Myc may inactivate p53 through MILIP in cancer cells. Substantiating this, siRNA silencing of c-Myc or treatment with the small-molecule c-Myc inhibitor 10058-F4 upregulated p53, which was nevertheless abolished by overexpression of MILIP (Fig. 1g, Supplementary Fig. 5n). In contrast, c-Myc overexpression caused reduction in p53 expression that was diminished by silencing of MILIP (Fig. 1h). Collectively, these results demonstrate that MILIP links c-Myc to inactivating p53. This regulation appears independent of p14[ARF] as the cell lines used (A549, MCF-7, HCT116, MDA-MB-231, and U2OS) were deficient in p14[ARF] expression[37–43].

**MILIP promotes tumourigenicity.** SiRNA silencing of MILIP inhibited cell viability in diverse types of cancer cell lines (Supplementary Fig. 6a–c). Moreover, induced shRNA silencing of MILIP retarded the tumourigenicity of A549 and MCF-7 cells in vitro and in A549 xenograft models (Fig. 2a, b, Supplementary Fig. 6d–f). In contrast, overexpression of MILIP promoted, albeit moderately, clonogenicity of A549 and MCF-7 cells, whereas cessation of induced MILIP silencing led to recovery of the growth of A549 xenografts along with MILIP expression (Fig. 2b, Supplementary Fig. 6e–g). Silencing of MILIP triggered apoptosis and arrest of cell cycle progression at G0/G1 phase (Supplementary Fig. 7a–c), both functional characteristics of p53 activation and consistent with the anticipated actions of MILIP against p53[44,45]. Accordingly, co-knockdown of p53 reversed the inhibitory effect of MILIP knockdown on clonogenicity (Fig. 2c, Supplementary Fig. 7d), reflecting that MILIP expression is integral for sustaining cancer cell survival and division through repressing p53. We confirmed the suppressive effect of MILIP inhibition on A549 cell clonogenicity using CRISPRi silencing of MILIP (Supplementary Fig. 7e). Akin to c-Myc, high MILIP

expression was associated with poor overall survival (OS) in various cancer types (Fig. 2d, Supplementary Fig. 8a), supporting the notion that MILIP upregulation contributes to c-Myc-driven cancer maintenance and progression[46,47].

Of note, MILIP levels did not differ among tumors of different stages (Supplementary Fig. 8b), suggesting that MILIP upregulation is an early event during c-Myc-driven tumourigenesis[48]. Indeed, MILIP expression was increased in pre-neoplastic colon lesions (adenomas) compared with normal colon epithelia (Fig. 2e, Supplementary Fig. 8c). Silencing of MILIP did not influence the viability of MCF10A and HME-1 human mammary epithelial cells (Supplementary Fig. 8d), suggesting that MILIP did not have a major role in regulating normal cell survival and proliferation. However, silencing of MILIP decelerated anchorage-independent growth of HME-1 caused by c-Myc overexpression in conjunction with knockdown of p14[ARF] (Fig. 2f, Supplementary Fig. 8e, f). Similarly, silencing of MILIP retarded c-Myc-driven anchorage-independent growth of p14[ARF]-negative MCF10A cells (Fig. 2g, Supplementary Fig. 8g), whereas conversely, MILIP overexpression promoted their growth (Fig. 2h, Supplementary Fig. 8h). Thus, MILIP upregulation contributes to c-Myc-driven neoplastic transformation.

**MILIP binds to and promotes p53 polyubiquitination and degradation.** We returned to investigate the mechanism responsible for MILIP-mediated inactivation of p53. While MILIP did not impinge on p53 mRNA expression (Supplementary Fig. 9a), silencing of MILIP prolonged the half-life time of p53, which was associated with reduced polyubiquitination of the protein (Fig. 3a, b, Supplementary Fig. 9b). In contrast, overexpression of MILIP increased p53 polyubiquitination and reduced its expression (Supplementary Fig. 9c, d). This reduction in p53 expression was abolished by the addition of the proteasomal inhibitor MG132 (Supplementary Fig. 9d), confirming that MILIP inactivates p53 through promoting its polyubiquitination and subsequent proteasomal degradation. Indeed, silencing of MDM2, the major E3 ubiquitin–protein ligase responsible for polyubiquitinating p53[49], diminished the decrease in p53 expression caused by MILIP overexpression (Supplementary Fig. 9e). However, the association between MDM2 and p53 was not reduced by MILIP knockdown (Fig. 3c), suggesting that this is not due to alterations in the binding between MDM2 and p53. Of note, knockdown of MAFG did not alter the half-life time of p53 protein in A549 cells (Supplementary Fig. 9f), ruling out the possibility that MAFG contributes to the regulation of p53 expression.

Strikingly, MILIP appeared to be an RNA binding partner of p53, as it physically associated with p53 in A549 and MCF-7 cells

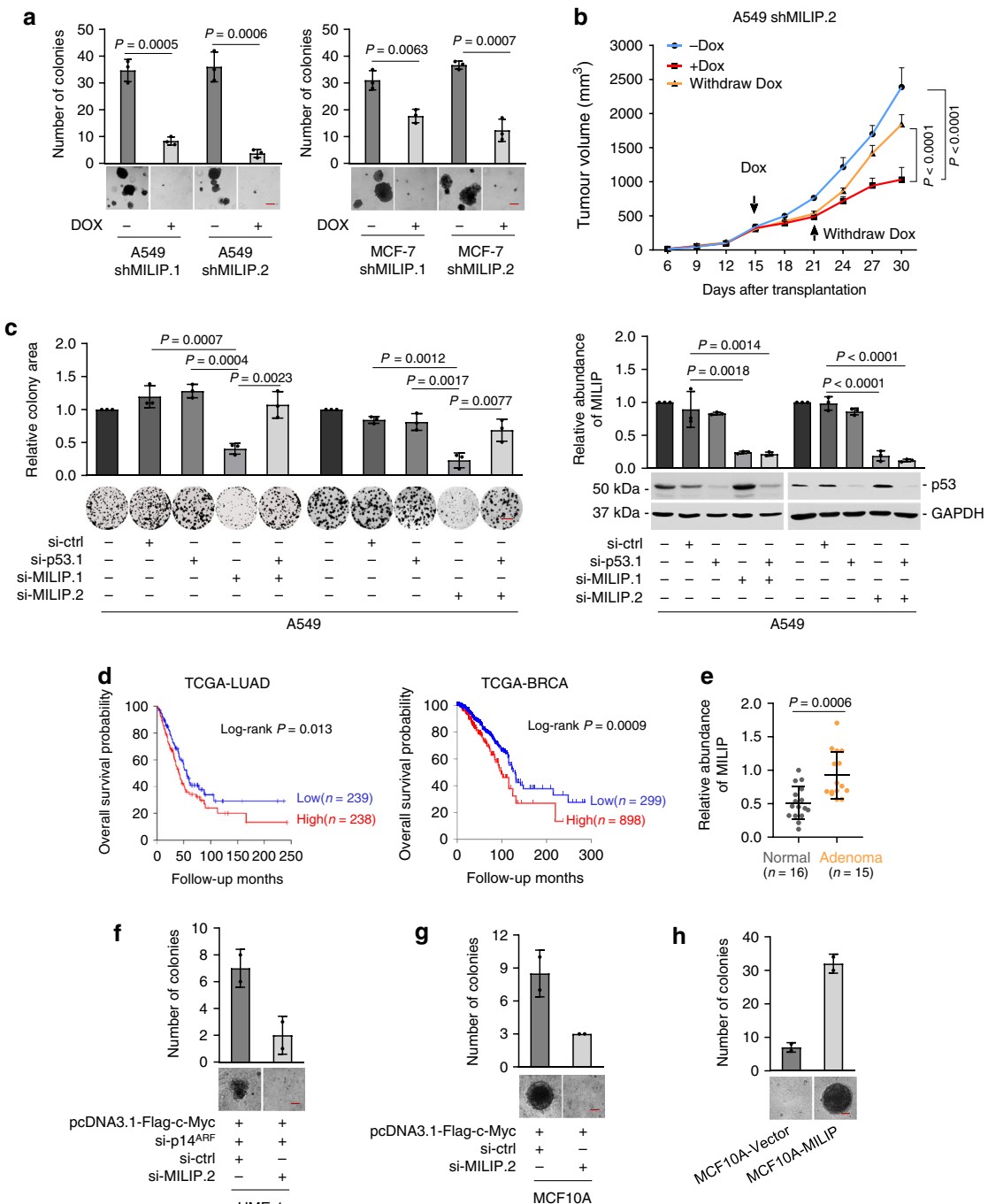

**Fig. 2 MILIP promotes tumorgenicity. a** Induced knockdown of MILIP by DOX reduced cancer cell anchorage-independent growth. Scale bar, 50 μm. Data are representatives or mean ± s.d.; n = 3 independent experiments, two-tailed Student's t test. DOX: 200 ng/mL. **b** Induced knockdown of MILIP by DOX retarded A549 xenograft growth, which was reversed by DOX withdrawal in nu/nu mice. Data are mean ± s.d.; n = 6 mice per group, one-way ANOVA followed by Tukey's multiple comparison test. DOX: 2 mg/mL supplemented with 10 mg/mL sucrose in drinking water. **c** MILIP silencing reduced A549 cell clonogenicity, which was attenuated by co-silencing of p53 siRNA1. Scale bar, 1 cm. Data are representatives or mean ± s.d.; n = 3 independent experiments, one-way ANOVA followed by Tukey's multiple comparison test. **d** Kaplan–Meier analysis of the probability of overall survival (OS) of lung adenocarcinoma (LUAD; n = 477) and breast invasive carcinoma (BRCA; n = 1197) derived from the TCGA using the median/quartile of MILIP levels as the cut-off. **e** MILIP expression was increased in colon adenomas (n = 15) compared with normal colon epithelia (n = 16). Data are mean ± s.d.; two-tailed Student's t test. **f** MILIP silencing decelerated anchorage-independent growth of HME-1 human mammary epithelial cells caused by c-Myc overexpression along with knockdown of p14$^{ARF}$. Scale bar, 50 μm. Data are representatives or mean ± s.d.; n = 2 independent experiments. **g** MILIP silencing decelerated anchorage-independent growth of MCF10A human mammary epithelial cells that did not express p14$^{ARF}$ caused by c-Myc overexpression. Scale bar, 50 μm. Data are representatives or mean ± s.d.; n = 2 independent experiments. **h** MILIP overexpression caused anchorage-independent growth of MCF10A cells. Scale bar, 50 μm. Data are representatives or mean ± s.d.; n = 2 independent experiments. Source data of Fig. 2a–h are provided as a Source Data file.

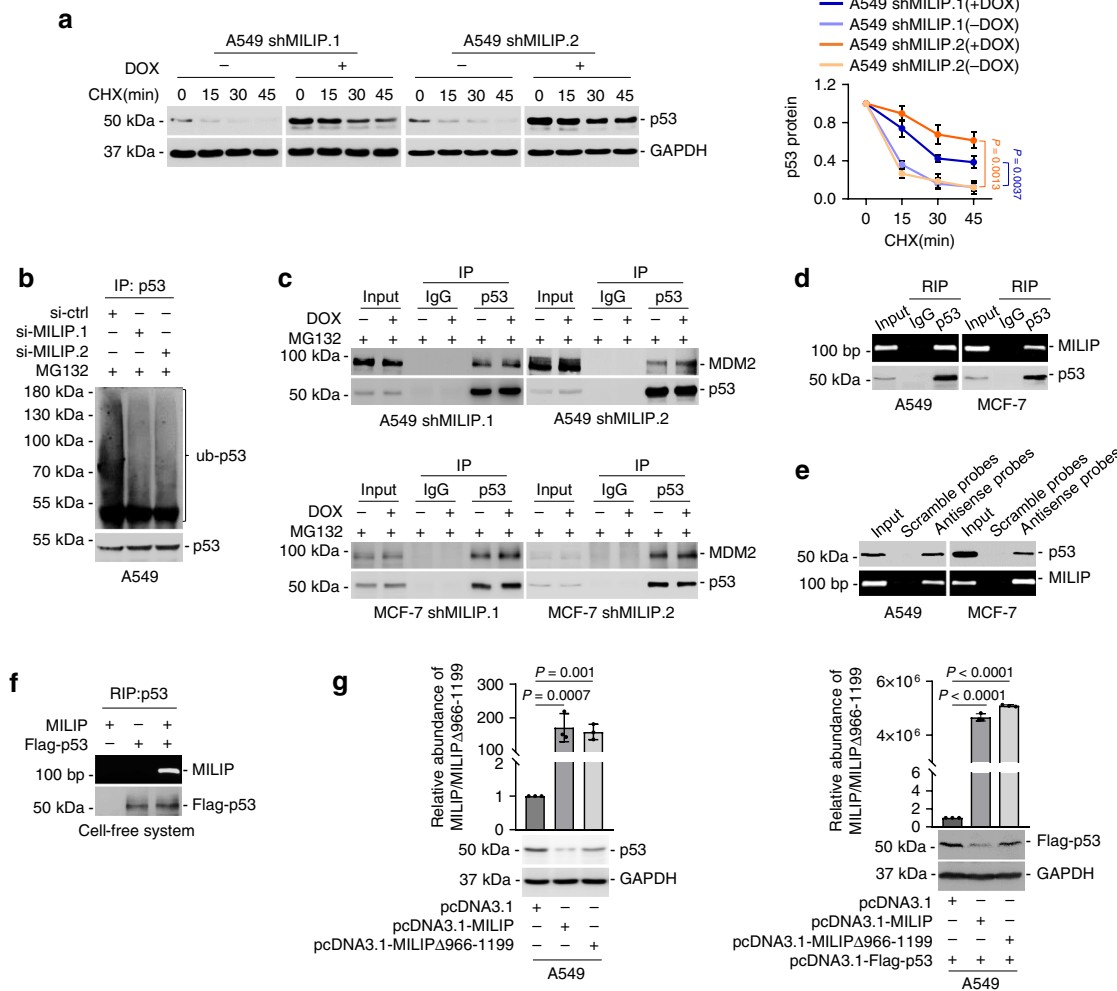

**Fig. 3 MILIP binds to and destabilizes p53. a** Induced knockdown of MILIP prolonged the half-life time of p53 protein in cycloheximide (CHX)-chase assays. Data are representatives or mean ± s.d.; $n = 3$ independent experiments, two-tailed Student's $t$ test. CHX: 40 μg/mL, DOX: 200 ng/mL. **b** MILIP silencing reduced p53 polyubiquitination. Data shown represent three independent experiments. **c** Induced knockdown of MILIP did not reduce the binding between p53 and MDM2. Data shown represent three independent experiments. **d** MILIP was coprecipitated with p53 using RNA immunoprecipitation (RIP) assays. Data shown represent three independent experiments. **e** p53 was co-pulled down with MILIP by antisense probes against MILIP using RNA pulldown assays. Data shown represent three independent experiments. **f** In vitro transcribed MILIP was coprecipitated with purified Flag-tagged p53 in a cell-free system using RNA immunoprecipitation (RIP) assays. Data shown represent three independent experiments. **g** Overexpression of MILIP but not a MILIP mutant with the −966/−1199 segment deleted (MILIP Δ−966/−1199) downregulated both endogenous (left) and overexpressed exogenous (right) p53 protein levels. Data are mean ± s.d.; $n = 3$ independent experiments, one-way ANOVA followed by Tukey's multiple comparisons test. Source data of Fig. 3a–g are provided as a Source Data file.

and bound to purified p53 in a cell-free system (Fig. 3d–f, Supplementary Fig. 10a)[50]. This was supported by the colocalization of p53 and MILIP as shown in studies using immunofluorescent staining of p53 and fluorescence in situ hybridization (FISH) analysis of MILIP in cells grown on coverslips (Supplementary Fig. 10b). Deletion mapping revealed that MILIP bound to p53 through its E2 (Supplementary Fig. 10c, d). Further deletion of randomly defined fragments of E2 showed that the −966/−1199 segment was critical for the binding of MILIP to p53 (Supplementary Fig. 10c, d). On the other hand, the C-terminus and the DNA-binding domain (DBD) of p53 were required for its interaction with MILIP (Supplementary Fig. 10e, f). Introduction of a MILIP mutant with the −966/−1199 segment deleted failed to downregulate p53 (Fig. 3g), demonstrating that the direct interaction is necessary for MILIP-mediated repression of p53.

Analysis of subcellular fractions by qPCR revealed that ~60–70% MILIP was cytoplasmic, whereas small amounts of MILIP located to the nucleus in A549 and MCF-7 cells (Supplementary Fig. 11a, b). Similarly, ISH analysis of cells grown in coverslips also showed cytoplasmic and nuclear localization of MILIP (Supplementary Fig. 3e). On the other hand, semiquantitative Western blot analyses showed that the vast majority of p53 was in the nucleus, whereas ~10% and ~4% of the protein was detected in the cytoplasm of A549 and MCF-7 cells, respectively (Supplementary Fig. 11c). Consistent with these varying distributions of MILIP and p53, RNA immunoprecipitation (RIP) assays showed that substantially higher amounts of MILIP were associated with p53 in the cytoplasm, whereas markedly lower amounts of MILIP were coprecipitated with p53 in the nuclear fractions (Supplementary Fig. 11d).

As a tumor suppressor, the abundance of p53 in fast-growing unstressed cancer cells is kept low[51]. However, past studies have documented wide variations in the absolute concentration of the protein within a cell, conceivably due to varying cell types and experimental approaches employed by different studies[52–54]. We

carried out absolute quantitation of p53 using the enzyme-linked immunosorbent assay (ELISA)[52]. The results showed that the number of p53 molecules per A549 and MCF-7 cell was 1422 and 2858, respectively (Supplementary Fig. 11e). Taking into account the distribution of MILIP and p53 in different subcellular compartments (Supplementary Fig. 11a, c), the relative level of MILIP vs. p53 was 109/142 ($\approx$1:1.3) and 65/86 ($\approx$1:1.3) in the cytoplasm of A549 and MCF-7 cells, respectively (Supplementary Fig. 11f). In contrast, the relative level ranged from 42/1280 ($\approx$1:30) to 47/2772 ($\approx$1:102) in the nucleus (Supplementary Fig. 11f). Depletion of MILIP through RNA pulldown abolished the presence of p53 in the cytoplasmic fractions (Supplementary Fig. 11g), indicating sufficient stoichiometric amounts of MILIP to interact with p53 in the cytoplasm. Given that p53 polyubiquitination and proteasomal degradation primarily occurs in the cytoplasm[55], these results suggest that the promoting effects of MILIP on p53 are largely accounted for by the cytoplasmic interaction of MILIP with p53.

**MILIP inhibition of p53 SUMOylation promotes its polyubiquitination.** We further investigated the mechanism of MILIP-mediated polyubiquitination and degradation of p53. Intriguingly, three of the top ten most upregulated genes following MILIP knockdown were TRIML2, NUPR1, and HDAC5, all known to interact with p53 and regulate p53 activity (Supplementary Fig. 12a)[56–58]. However, silencing of TRIML2 but not NUPR1 and HDAC5 reduced the expression of p53 in A549 cells (Supplementary Fig. 12b), proposing that TRIML2 may be involved in the regulation of p53 by MILIP. TRIML2 is a SUMO E3 ligase, which binds to and modifies p53 with SUMO-2/3[56]. Indeed, MILIP knockdown increased modification of p53 by SUMO-2/3, which was counteracted by co-knockdown of TRIML2, recapitulating the effect of treatment with the protein SUMOylation inhibitor ginkgolic acid (GA) (Fig. 4a, Supplementary Fig. 12c)[59]. Conversely, overexpression of MILIP antagonized increases in p53 SUMOylation resulting from overexpression of TRIML2 (Fig. 4b). Therefore, MILIP negatively regulates p53 SUMOylation through supressing TRIML2. Of note, treatment with GA diminished the reduction in p53 polyubiquitination and upregulation of p53 protein caused by MILIP silencing (Supplementary Fig. 12d, e). Similarly, TRIML2 silencing abolished the increase in p53 expression caused by silencing of MILIP (Fig. 4c). Moreover, MILIP silencing failed to alter p53 polyubiquitination in cells with TRIML2 silenced (Fig. 4d). In contrast, overexpression of TRIML2 reduced p53 polyubiquitination and upregulated its expression (Fig. 4e, f), similar to the effects of knockdown of MILIP (Fig. 3b, Supplementary Fig. 5b). As lysine 386 in p53 is the target for SUMO-2/3 and mutation of glutamic acid 388 abrogates SUMOylation while sparing K386 ubiquitination[19,60], we tested whether the E388A mutation affects MILIP-mediated promotion of p53 polyubiquitination. As expected, neither overexpression of TRIML2 nor MILIP knockdown affected SUMOylation of E388A mutant p53 (Supplementary Fig. 12f, g).

TRIML2 and p53 could be reciprocally co-immunoprecipitated in A549 and MCF-7 cell lysates (Supplementary Fig. 13a, b). However, overexpression of MILIP reduced their association (Supplementary Fig. 13c), whereas MILIP silencing increased the amount of TRIML2 co-immunoprecipitating with p53 (Fig. 4g). Deletion mapping experiments with p53 mutants showed that removing the p53 DBD but not other regions abolished its association with TRIML2 (Supplementary Figs. 10e and 13d). These results, along with those showing that the DBD of p53 is also required for its binding to MILIP (Supplementary Fig. 10e, f), suggesting that MILIP and TRIML2 can compete for binding to

p53. Supporting this notion, knockdown of TRIML2 resulted in increased binding of MILIP to p53 (Fig. 4h), whereas TRIML2 overexpression reduced the amount of MILIP associating with p53 (Supplementary Fig. 13e). Furthermore, TRIML2 localized predominantly to the cytoplasm and SUMO-2/3-modified p53 was readily detected in cytoplasmic but not nuclear fractions (Supplementary Fig. 13f, g), consistent with the involvement of MILIP in promoting p53 polyubiquitination and degradation in the cytoplasm. Of note, co-knockdown of p53 diminished the upregulation of TRIML2 caused by MILIP knockdown, indicating that upregulation of TRIML2 in cells with MILIP knockdown is primarily due to a p53-mediated increase (Supplementary Fig. 13h)[56]. Collectively these data establish that inhibition of TRIML2-mediated p53 SUMOylation by MILIP is necessary for its effects on p53 polyubiquitination.

Given the primary role of c-Myc in the regulation of MILIP, we also evaluated whether c-Myc exerts effects on p53 SUMOylation and polyubiquitination. Indeed, c-Myc knockdown stabilized p53 levels through enhancing its protein half-life time, which was associated with decreased polyubiquitination and concomitantly increased SUMOylation (Fig. 5a–c). Nevertheless, the stabilization and post-translational modification of p53 resulting from c-Myc depletion were reversed by overexpression of MILIP (Fig. 5d–f). Therefore, c-Myc-driven MILIP expression regulates p53 SUMOylation and thus promotes its polyubiquitination and proteasomal degradation (Supplementary Fig. 14).

In summary, we have demonstrated that the pan-cancer lncRNA MILIP links c-Myc to repression of the tumor suppressor p53 (Supplementary Fig. 14). As a proto-oncoprotein, c-Myc is cast as an enigmatic actor, playing dualistic roles as both villain and hero. For the latter, c-Myc triggers activation of p53 through p14$^{ARF}$, which serves as a key checkpoint to curb malignant transformation through induction of apoptosis[11,61]. Here, we establish that c-Myc inactivates p53 through MILIP, providing an explanation as to how wild-type p53 can be repressed by c-Myc independently of the loss of p14$^{ARF}$ (Supplementary Fig. 14). Noticeably, p14$^{ARF}$ mRNA but not MILIP levels correlate with *MYC* gene expression in normal human tissues (Supplementary Fig. 15)[62], whereas MILIP is upregulated and its expression is positively associated with the levels of *MYC* expression in human cancers (Fig. 1d, e, Supplementary Figs. 1b, 2k, and 3a, c, d). It is thus conceivable that transcriptional activation of *MILIP* requires oncogenic (relatively high) levels of c-Myc, whereas physiological (relatively low) levels of c-Myc is sufficient for transcriptional activation of p14$^{ARF}$ (47).

Through bioinformatics analysis, we identified a transcript in Pan troglodytes that is highly homologous to human MILIP with 93% sequence similarity (Supplementary Table 3), suggesting evolutionary conservation of MILIP between primate species. However, the lack of similarity between human MILIP and Mus musculus transcripts makes it infeasible to test the role of MILIP in cancer initiation in transgenic mouse models (Supplementary Table 3). Nevertheless, our results from studies using human cell line models and pre-malignant tissues suggest that MILIP may contribute to c-Myc-driven neoplastic transformation, and MILIP may therefore represent a potential anti-cancer target for counteracting the c-Myc-axis.

## Methods
**Cell culture and human tissues.** Human lung cancer A549 cells, human breast cancer MCF-7 and MDA-MB-231 cells, human colon cancer HCT116 cells were maintained in DMEM supplemented with 10% fetal bovine serum (FBS) and 1% penicillin–streptomycin. Human osteosarcoma U2OS cells were cultured in MacCoy's 5 A with 10% FBS and 1% penicillin–streptomycin. Human B-cell lymphoma P493-6 cells were cultured in RPMI-1640 with 10% FBS and 1% penicillin–streptomycin. The nontumourigenic mammary epithelial MCF10A cells were cultured in DMEM/F12 medium supplemented with 5% horse

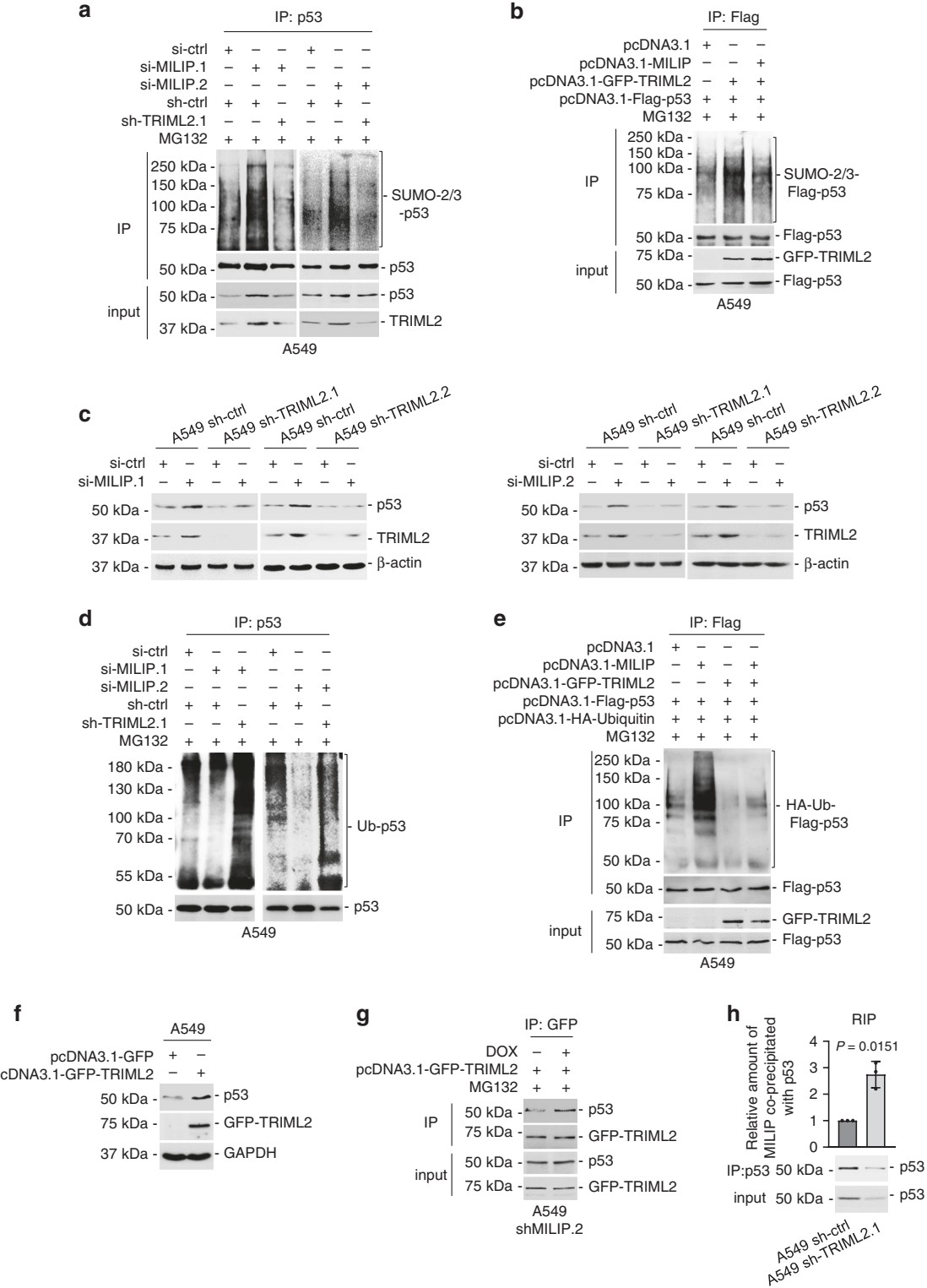

**Fig. 4 MILIP promotes p53 polyubiquitination through supressing TRIML2-mediated p53 SUMOylation. a** Knockdown of MILIP increased modification of p53 by SUMO-2/3, which was abolished by co-knockdown of TRIML2. Data shown represent three independent experiments. MG132: 10 μM. **b** TRIML2 overexpression increased p53 SUMOylation, which was diminished by co-overexpression of MILIP. Data shown represent three independent experiments. MG132: 10 μM. **c** Knockdown of MILIP upregulated p53, which was diminished by co-knockdown of TRIML2. Data shown represent three independent experiments. **d** Knockdown of MILIP-reduced p53 polyubiquitination, which was reversed by TRIML2 co-knockdown. Data shown represent three independent experiments. MG132: 10 μM. **e** MILIP overexpression increased p53 polyubiquitination, which was abolished by overexpression of TRIML2. Data shown represent three independent experiments. MG132: 10 μM. **f** Overexpression of TRIML2 upregulated p53. Data shown represent three independent experiments. **g** Knockdown of MILIP increased the amount of TRIML2 bound to p53. Data shown represent three independent experiments. IP immunoprecipitation. The proteasome inhibitor MG132 was used at 10 μM. **h** Knockdown of TRIML2 increased the amount of MILIP bound to p53. Data are representatives or mean ± s.d.; n = 3 independent experiments, two-tailed Student's t test. Source data of Fig. 4a–h are provided as a Source Data file.

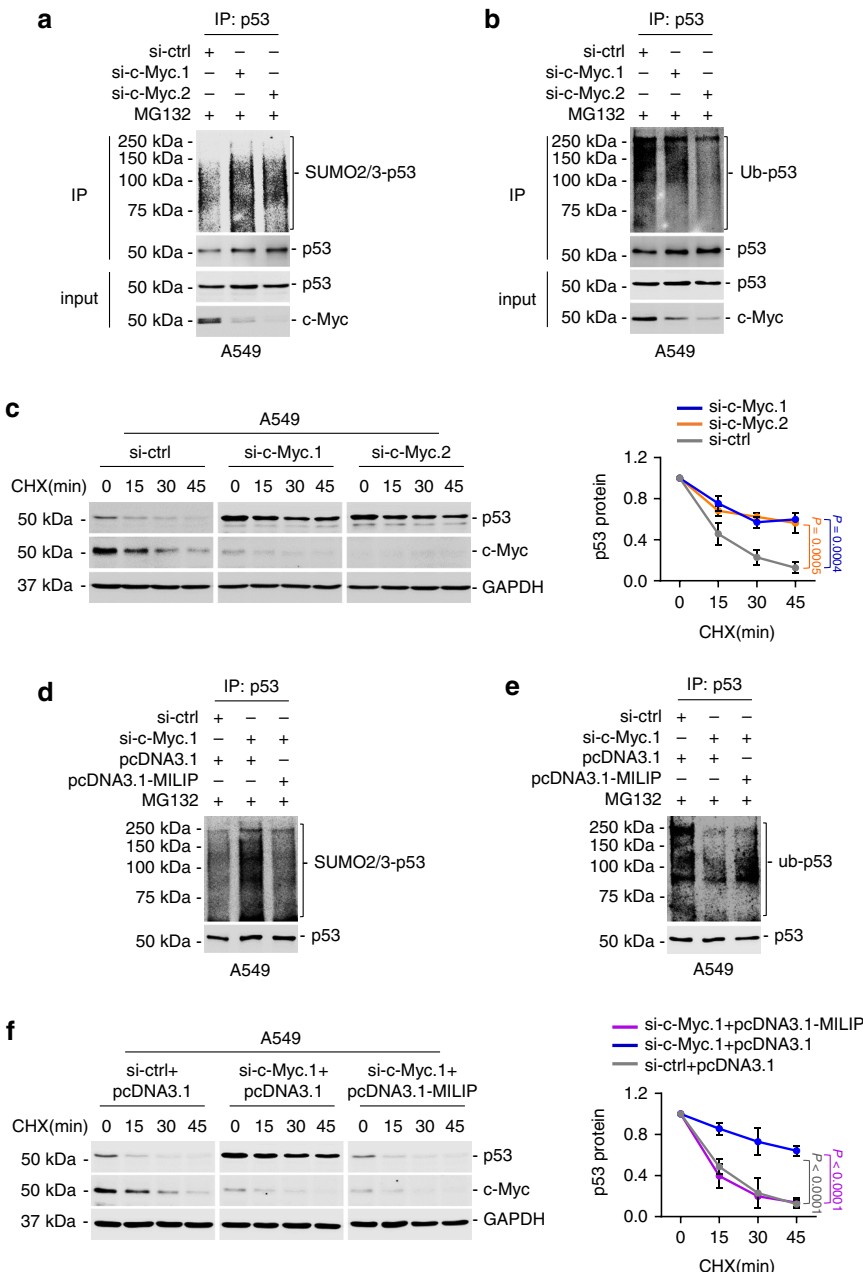

**Fig. 5 c-Myc represses p53 through MILIP. a, d** Knockdown of c-Myc increased modification of p53 by SUMO-2/3, which was reversed by overexpression of MILIP. Data shown represent three independent experiments. MG132: 10 μM. **b, e** Knockdown of c-Myc decreased p53 polyubiquitination, which was reversed by overexpression of MILIP. Data shown represent three independent experiments. MG132: 10 μM. **c, f** Knockdown of c-Myc prolonged the half-life time of p53 protein, which was reversed by overexpression of MILIP in CHX-chase assays. Data are representatives or mean ± s.d.; $n = 3$ independent experiments, one-way ANOVA followed by Tukey's multiple comparisons test. CHX: 40 μg/mL. Source data of Fig. 5a–f are provided as a Source Data file.

serum, 10 μg/mL insulin, 0.5 μg/mL hydrocortisone, 100 ng/mL Cholera toxin and 20 ng/mL EGF. Normal human breast epithelial HME-1 cells were maintained in DMEM/F12 medium supplemented with 10% FBS, 10 μg/mL insulin, 0.5 μg/mL hydrocortisone, 10 ng/mL EGF. Cells were cultured in a humidified incubator at 37 °C and 5% $CO_2$ and were tested using RT-PCR for mycoplasma contamination. Individual cell line authentication was confirmed using the AmpFlSTR Identifiler PCR Amplification Kit from Applied Biosystems and GeneMarker V1.91 software (SoftGenetics LLC)[63].

Formalin-fixed paraffin-embedded colon cancer and lung cancer tissue microarrays were purchased from the Shanghai Outdo Biotech Co., Ltd. (Cat. HColA180Su12, Cat. HLugA180Su03). Freshly removed colon adenoma tissues and adjacent normal colon tissues were obtained from patients undergoing surgical resection at the Department of General Surgery at Shanxi Cancer Hospital. Studies using human tissues were approved by the Human Research Ethics Committees of the Shanxi Cancer Hospital in agreement with the guidelines set forth by the Declaration

of Helsinki. The study is compliant with all relevant ethical regulations for Human research participants and all participants provided written informed consent.

**LncRNA upregulation pattern analysis in cancers.** To evaluate the statistical significance of upregulated lncRNAs among 20 cancer types, we analyzed the expression level of 3136 lncRNAs in 7584 cancer samples compared with corresponding normal tissues in TCGA, downloaded from Cancer RNA-seq Nexus dataset. The lncRNA list was ranked according to percentage of upregulated cancer cases in descending order. Upregulation in cancer was defined where the lncRNA expression value in cancer versus the mean lncRNA expression value in corresponding normal tissue was greater than 2× standard deviation.

**Antibodies and reagents.** Information on antibodies used in this study is provided in Supplementary Table 4. The following reagents were purchased from indicated

companies: 10058-F4 (Sigma, F3680), MG132 (Sigma, M7449), doxycycline (Sigma, D9891), GA C15:1 (Sigma, 75741), N-Ethylmaleimide (Sigma, 04259), Propidium iodide (Sigma, P4170), Cycloheximide (MP Biomedicals, 100183).

**Quantitative PCR (qPCR).** Total cellular RNA isolated using an ISOLATE II RNA Mini Kit (Bioline) was subjected to PCR analysis. The $2^{-\Delta\Delta CT}$ method was used to calculate the relative gene expression levels in comparison to the GAPDH or 18S rRNA housekeeping controls. Primer sequences are listed in Supplementary Table 5.

**In vitro transcription.** The plasmid pcDNA3.1-MILIP was constructed by Tolo Biotechnology. The plasmids were linearized by restriction enzyme BstBI (New England biolab) and in vitro transcription was then performed using TranscriptAid T7 High Yield Transcription Kit (Thermo Fisher Scientific) according to the manufacturer's instructions.

**Inducible shRNA knockdown.** ShRNA sequences were constructed into FH1-tUTG inducible knockdown vector. The lentiviral particles were packaged via co-transfection with FH1-tUTG (44 μg), pMDLg.pRRE (22 μg), pMD2.g (13.2 μg), and pRSU.pREV (11 μg) plasmids into HEK293 cells in a T175 culture flask[33,64]. A549 or MCF-7 inducible knockdown cell sublines were established after the lentiviral transduction. shRNA sequences are shown in Supplementary Table 6.

**CRISPR interference (CRISPRi).** Lentiviral particles carrying dCas9-KRAB and plasmids pCRISPR-LvSG03 containing scramble or MILIP sgRNA were purchased from GeneCopoeia. Lentiviral particles were packaged in HEK293 cells via co-transfection with pCRISPR-LvSG03-sgRNA (44 μg), pMDLg.pRRE (22 μg), pMD2.g (13.2 μg), and pRSU.pREV (11 μg) plasmids[33,64]. A549 cells were first transduced with dCas9-KRAB and selected with blasticidin before transduction with scramble or MILIP sgRNA and further selection using puromycin. The MILIP sgRNA sequence is listed in Supplementary Table 6.

**CRISPR/Cas9.** Two sgRNAs targeting the c-Myc-BR in the MILIP promoter were constructed in the lentiCRISPRv2 vector (Addgene, Cat#52961) before transfection and selection using puromycin (2 μg/mL). Genomic DNA was extracted from single cell clones using the Wizard® SV Genomic DNA Purification System (Promega, Cat#A2361) and genomic DNA flanking the CRISPR-targeted region was amplified by PCR using the AmpliTaq Gold™ 360 Master Mix (Applied Biosystems, Cat#4398881). PCR products were purified using the Isolate II PCR and Gel Kit (Bioline, Cat#BIO-52060) and analyzed by Sanger sequencing. The c-Myc-BR sgRNA sequences and PCR primers used are listed in Supplementary Tables 6 and 7, respectively.

**Apoptosis.** Apoptotic cells were quantitated using the FITC Annexin V Apoptosis Detection Kit (BD Biosciences, Cat#556547) according to the manufacturer's instructions. Briefly, cells were resuspended in binding buffer and incubated with Annexin V/propidium iodide (PI) for 15 min at room temperature in dark before analysis using a flow cytometer (FACSCanto II, BD Biosciences).

**Absolute quantification of MILIP.** Absolute RNA quantification was performed using the standard curve method. cDNA was prepared from a fixed cell number using the qScript cDNA SuperMix (Quantabio, Cat#95048-500) in a 20 μL reaction and subsequently diluted to 100 μL. Tenfold serial dilutions of pcDNA3.1-MILIP plasmid were used to construct standard curves. Assays were reconstituted to a final volume of 20 μL using 5 μL cDNA/standard and cycled using a 7900HT Fast Real-Time PCR System. Data calculated as copies per 5 μL cDNA were converted to copies per cell based on the known input cell equivalents. Primer sequences used are listed in Supplementary Table 5.

**Absolute quantification of p53 protein.** P53 protein concentrations were measured using PathScan Total p53 Sandwich ELISA Kit (Cell Signaling Technology, Cat#7370C), and Recombinant Human TP53 protein (Creative Biomart, Cat#TP53-15H) was used as a standard. P53 protein copies per cell were calculated according to p53 concentrations and the cell equivalents of lysate.

**Subcellular fractionation.** Cells were incubated with hypotonic buffer A (10 mM Hepes pH 7.9, 10 mM KCl, 0.1 mM EDTA, 0.1 mM EGTA, 1 mM DTT, 0.15% Triton X-100, cOmplete™, EDTA-free Protease Inhibitor Cocktail) and swollen on ice for 15 min. Samples were centrifuged for 3 min at 12,000 × g, and the supernatant collected as the cytoplasmic fraction. The pellets were rinsed once with cold PBS and nuclear proteins were extracted using an equal volume of buffer B (20 mM Hepes pH 7.9, 400 mM NaCl, 1 mM EDTA, 1 mM EGTA, 1 mM DTT, 0.5% Triton X-100, cOmplete™, EDTA-free Protease Inhibitor Cocktail) on ice for 15 min. Cytoplasmic and nuclear fraction were centrifuged at 16,000 × g for 20 min to remove insoluble debris.

**Immunofluorescence (IF).** Cells grown on coverslips were fixed for 10 min (4% formaldehyde in DEPC-treated PBS) at room temperature, washed using PBS and then permeabilized using blocking buffer for 60 min at room temperature. Cells were then incubated overnight at 4 °C with p53 antibody diluted 1:200 in blocking buffer (Protein Tech, 10442-1-AP) before washing three times with PBS and then detection with Alexa Fluor 488-goat anti rabbit secondary antibody diluted 1:1000 in blocking buffer (Thermo Fisher Scientific, A11034) at room temperature for 60 min in the dark. Cells were washed again in PBS and fixed before further analysis.

**RNA fluorescence in situ hybridization (RNA FISH).** RNA FISH was performed using Cy3-MILIP probes synthesized by Sangon Biotech[65]. Briefly, cells cultured on coverslips were incubated with probes (2 μg/mL) diluted in hybridization buffer at 37 °C overnight after being equilibrated in wash buffer for 5 min at room temperature. Cells were then washed twice with wash buffer for 20 min each at 37 °C and mounted in Antifade Mountant with NucBlue reagent (Thermo Fisher Scientific, P36981). Images were digitally recorded using Leica SP8 confocal microscope. The Cy3-MILIP probe sequences are listed in Supplementary Table 7.

**In situ hybridization (ISH).** ISH assays were performed using the RNAscope® 2.0 HD detection kit-Brown (Advanced Cell Diagnostics, Hayward, CA) according to the manufacturer's instructions[66]. Briefly, FFPE tissue sections (4 μm thick) were deparaffinized and rehydrated, then heated and treated with proteinase K. Sections were then hybridized with probes at 40 °C for 3 h. After washing, the sections were incubated with 3,3'-diaminobenzidine (DAB), and counterstaining was carried out using hematoxylin. Positive staining was identified as brown, punctate dots present in cells.

The percentage of positive cells was estimated from 0 to 100%. Intensity of staining (intensity score) was judged on an arbitrary scale of 0–4: no staining (0), weakly positive staining (1), moderately positive staining (2), strongly positive staining (3), and very strong positive staining (4). A reactive score (RS) was derived by multiplying the percentage of positive cells with staining intensity divided by 10.

**Immunoprecipitation (IP).** Cells were lysed with lysis buffer (20 mM Tris-HCl pH 8.6, 100 mM NaCl, 20 mM KCl, 1.5 mM MgCl₂, 0.5% NP-40, cOmplete™ EDTA-free Protease Inhibitor Cocktail) on ice for 1 h. Samples were clarified by centrifugation at 16,000 × g for 30 min, and the supernatants incubated with the specified antibodies at 4 °C overnight. Protein–antibody complexes were then captured with protein A/G agarose beads (Life Technologies, 20421) at 4 °C for 2 h mixing by rotation and the beads then rinsed with wash buffer (25 mM Tris, 150 mM NaCl, pH 7.2), boiled and subjected to immunoblotting analysis. Ubiquitination assays were performed using cell lysates that were first boiled and sonicated[67,68]. For detection of p53 SUMOylation, 50 mM N-ethylmaleimide (NEM) (Sigma, E04259) was added to the lysis buffer before lysing the cells.

**Biotin RNA pulldown (RPD).** Cell lysates were prepared by ultrasonication in lysis buffer (50 mM Tris-HCl [pH 7.5], 150 mM NaCl, 2.5 mM MgCl₂, 1 mM EDTA, 10% Glycerol, 0.5% Nonidet P-40/Igepal CA-630, 1 mM DTT, cOmplete™ EDTA-free Protease Inhibitor Cocktail and RNase inhibitors). Probes were incubated with lysate at 4 °C overnight before rotating with streptavidin beads (Invitrogen) for additional 2 h. Then beads were washed in lysis buffer for four times. Probe sequences are shown in Supplementary Table 7.

**RNA immunoprecipitation.** RIP was performed with an EZ-Magna RIP Kit (17-701; Millipore) according to the instructions provided by the manufacturer. Briefly, approximately $2 \times 10^7$ cells were lysed in hypotonic buffer supplemented with RNase inhibitor and protease inhibitor before centrifugation. Cell lysates were incubated with magnetic beads coated with the indicated antibodies at 4 °C for 4 h to overnight. After extensive washing using RIP wash buffer, the bead-bound immunocomplexes were treated with proteinase K at 55 °C for 30 min. To isolate RNAs, samples were centrifuged and placed on a magnetic separator, and supernatants were used to extract RNA by an ISOLATE II RNA Mini Kit (Bioline). Purified RNAs were then subjected to PCR analysis.

**Chromatin immunoprecipitation.** The ChIP assays were performed by using the MAGnify™ Chromatin Immunoprecipitation System (Thermo Fisher, 492024) according to the manufacturer's instructions. The bound DNA fragments were subjected to PCR using the specific primers. Primers used in this study are shown in Supplementary Table 7[69].

**Luciferase reporter assays.** Assays were performed according to the manufacturer's instructions (Promega). Cells were transfected with the pGL3-based constructs containing *MILIP* promoter together with Renilla luciferase plasmids. Twenty-four hours later, firefly and Renilla luciferase activities were examined by Dual-Luciferase® Reporter Assay System (Promega) and Renilla luciferase activities were used to normalize the firefly luciferase activity.

**Cell viability assay**. Briefly, cells were seeded at $5 \times 10^3$/well in 96-well plates overnight before experimental treatments. WST-8 solution (10 μL) was added and incubated at 37 °C for 2 h. The absorbance at 450 nm was then recorded by Synergy 2 multidetection microplate reader (BioTek).

**Cell cycle analysis**. Cells were fixed by 70% ethanol on ice for 1 h and spun down at $1500 \times g$. Cell pellets were re-suspended in PBS containing 0.25% Triton X-100 and incubate on ice for 15 min. After discarding the supernatant, the cell pellet was resuspended in 0.5 ml PBS containing 10 μg/mL RNase A and 20 μg/mL PI stock solution and incubate at room temperature (RT) in the dark for 30 min. Cells were then subjected to analysis using a flow cytometer (FACSCanto, BD Biosciences)[70]. The gate strategy is shown in Supplementary Fig. 16.

**Colony formation**. Totally, $1 \times 10^3$ cells were seeded and incubated in a 6-well plate. Around two weeks later, cells were fixed, stained with crystal violet, and photographed.

**Anchorage-independent cell growth**. Totally, $1 \times 10^5$ cells were cultured in 6-well culture plates using a two-layer agar system (0.3% agar plating layer on top of 0.4% base agar layer). Cells were incubated at 37 °C in humidified incubator for further 30 days feeding with cell culture medium twice a week. Cell colonies were counted under a light microscope.

**Xenograft mouse model**. Cells expressing inducible MILIP shRNAs were subcutaneously injected into the dorsal flanks of 4-week-old female nude mice (6 mice per group, Shanghai SLAC Laboratory Animal Co. Ltd.). Mice were sacrificed, and tumors were excised and measured. Studies on animals were conducted in accordance with relevant guidelines and regulations and were approved by the Animal Research Ethics Committee of the first affiliated hospital, Shanxi Medical University and Shanxi Cancer Hospital and Institute. All mice were housed in a temperature-controlled room (21–23 °C) with 40–60% humidity and a light/dark cycle of 12 h/12 h.

**Statistical analysis**. Statistical analysis was carried out using Microsoft Excel software and GraphPad Prism to assess differences between experimental groups. Statistical differences were analyzed by two-tailed Student's $t$ test or one-way ANOVA test followed by Tukey's multiple comparisons. $P$ values lower than 0.05 were considered to be statistically significant.

**Reporting summary**. Further information on research design is available in the Nature Research Reporting Summary linked to this article.

## Data availability

The RNA sequencing data have been deposited in the NCBI Gene Expression Omnibus database under the accession code GSE141886. The long noncoding RNA expression data referenced during the study are available in a public repository from the Cancer RNA-seq Nexus dataset (http://syslab4.nchu.edu.tw/). MILIP expression data in adenoma and normal tissues referenced during the study are available in a public repository from the R2 website (https://hgserver1.amc.nl/cgi-bin/r2/main.cgi) under the accession codes Mixed Colon-Marra (GSE8671), Mixed Colon-Balazs (GSE4183), and Mixed Colon-Skrzypczak (GSE20916). The linear regression data referenced during the study are available in a public repository from the R2 website (https://hgserver1.amc.nl/cgi-bin/r2/main.cgi) under the accession codes TCGA-Glioblastoma, Disease Colon-Watanabe (GSE3629), Tumor Breast-Black (GSE36771), and Normal Tissues/Cells-Tsunoda (GSE18674). The cancer patient survival data referenced during the study are available in a public repository from the GEPIA website (http://gepia.cancer-pku.cn/) and the OncoLnc website (http://www.oncolnc.org/) under the accession codes TCGA-LUAD, TCGA-BRCA, TCGA-BLCA, TCGA-LIHC, and TCGA-KIRP. The source data underlying Figs. 1a–c, e–h, 2a–h, 3a–g, 4a–h, 5a–f, and Supplementary Figs. 1b, 2b–h, j–m, 3a, b, d, 4b–g, i, 5a–n, 6a, b, e–g, 7a–d, 8a–e, h, 9a–f, 10a, d, f, 11a–g, 12a–g, 13a–h, and 15 are provided as a Source Data file. All the other data supporting the findings of this study are available within the article and its supplementary information files and from the corresponding author upon reasonable request. A reporting summary for this article is available as a Supplementary Information file. Source data are provided with this paper.

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

## Acknowledgements

This work was supported by the National Health and Medical Research Council (NHMRC; APP1147271, APP1162753, and APP1177087) and Cancer Council NSW Project Grant (RG20-10), Australia, and National Natural Science Foundation of China (81972840). The authors thank A/Professor M. J. Herold (Walter and Eliza Hall Institute of Medical Research, Australia) for the plasmid FH1tUTG, Professor Mian Wu and Dr. Wanglai Hu (Translational Research Institute, Henan Provincial People's Hospital, China) for plasmids PG13-luc (wt p53 binding sites), pcDNA3.1-Flag-p53, pcDNA3.1-Flag-p53-ΔTAD, pcDNA3.1-Flag-p53-ΔDBD, pcDNA3.1-Flag-p53-ΔCTD, and pcDNA3.1-Flag-c-Myc, A/Professors Nikki Verrills and Pradeep Tanwar (School of Biomedical Sciences and Pharmacy, The University of Newcastle, Australia) for MCF10A cells and technical assistance and discussion, respectively.

## Author contributions

Y.C.F., L.J., and X.D.Z. designed the experiments. X.D.Z. and L.J. supervised the work. L.T. and S.T.G. performed the experiments using human tissues; Y.W., W.G., and S.T.G. conducted experiments in xenograft models; Y.C.F., X.Y.L., Q.J., Y.Y.Z., T. La, H.T., X.G.Y., M.F.J., and D.Z. conducted experiments using cell lines and related data collections; Y.C.F., J.M.L., L.J., and T. Liu carried out analysis of publicly available data and bioinformatics analysis. X.D.Z., R.F.T., R.J.S., and L.J. wrote the paper. All authors commented on the paper.

## Competing interests

The authors declare no competing interests.
