## [Peer Review File · Nature Communications]

Reviewers' comments:

Reviewer #1 (Remarks to the Author):

In this manuscript, the authors identified an interesting novel lncRNA MILIP. They showed that MILIP is upregulated in 18 different cancer types; MILIP is a downstream target of Myc, which includes the Myc binding region at its promoter. Knockdown of Myc abolished the expression of MILIP. From RNA-seq analysis in lung cancer cells, they also found that after knockdown of MILIP, the p53 signaling pathway is significantly upregulated. They also observed that knockdown of MILIP abolished cancer cell growth; meanwhile, lung cancer cell xenograft mouse model showed that mice with knockdown of MILIP had smaller tumors. Higher expression of MILIP also showed lower overall survival analyzed from the TCGA database. Mechanistically, authors identified that MILIP competed binding to p53 with one of p53 downstream target TRIML2 include specific binding site showed. Knockdown of MILIP downregulated expression of p53 and increasing the polyubiquitination of p53 and degradation. They also demonstrated overexpressed MILIP abolished the sumoylation that processed through TRIML2. Overall, it is an interesting manuscript. However, I have quite some concerns that need to be addressed.

Major concerns:

1. The main finding of this study is lncRNA MILIP. However, the sequence of MILIP and isoforms of MILIP are not reported. qRT-PCR analysis of MILIP expression and copy number in all the cell lines used should be listed. More structure information related to MILIP needs to be shown.
2. The localization of MILIP is very important, but it is not shown. From the extended figure 10a, authors seem to imply MILIP as a cytoplasm lncRNA. The localization needs to be clarified, and the location of sumoylation also needs to be identified.
3. At the beginning of the analysis, the rationale of choosing to study of MILIP is not clear.
4. Figure 1a, ChIP-PCR result is not enough; ChIP-seq should be performed and show the Myc binding region at the promoter.
5. In vitro study for the cell line choices are not clear. The authors identified MILIP as a pan-cancer lncRNA, but Figure 1 mostly focuses on lung and colon cancers. Later in figure 1h and figure 2, the authors move to breast cancer. Colon cancer cell line also need to be included in the later studies.
6. Previous studies showed that TRIML2 induced p53 is associated with apoptosis. The authors checked the caspase proteins but did not perform any apoptosis-related experiment.
7. In figure 2, most of the data showed by knockdown of MILIP, as an oncogene, overexpression data is also required to address the phenotype of this lncRNA. Similarly, the mouse model only showed knockdown MILIP xenograft, overexpressing mouse model is missing.
8. Extended figure 5a, authors showed overall survival probability in KIRC from TCGA, but this cancer type does not have upregulated MILIP expression shown in extended figure 1a, which lead to the confusion to this reviewer. Also, ACC is not included in extended figure 1a.
9. Figure 3c, the input expression of MDM2 in A549 cell lines are very different. The shMILIP.1 part showed a very high input expression of MDM2, but in the shMILIP.2 part, there is basically no input expression of MDM2. Also, the IgG sample in two trials are also different.
10. From the manuscript, knockdown MILIP decreased expression of p53 while upregulation of TRIML2 stabilize and induce p53 expression level. However, this phenomenon was not shown in several IP experiments include figure 4e IP and input p53 expression, 4h input p53 expression level, and extended figure 9e input and IP p53 expression.
11. The protein stability of p53 is not significantly induced after MILIP treatment in some figures.
12. What is the expression of MILIP in A549, MCF-7, HCT116, HME-1, MCF10A, and other (cancer) cell lines? MYC plays an important role in physiological conditions. Will MYC induce MILIP expression in normal tissue? What is MILIP's role in physiological condition?
13. MILIP is located to the same genomic locus as the protein-coding gene, MAFG. Does c-Myc regulate the transcriptional level of MAFG and MAFG also promotes the protein stability of p53?
14. Did the authors evaluate the effect of MILIP on neoplastic transformation with overexpression of MYC in MCF10A cells of Figure 2H? Only overexpression of MILIP promoting neoplastic

transformation shows MILIP could strongly regulate cell growth.

15. In Figure 3C, the authors have confirmed that knockdown of MILIP could significantly increase the protein stability of p53, but there is no significant induction after knockdown of MILIP in Input lanes. Similar results are also shown in Figure 4A (silencing of MILIP), 4B (overexpression of MILIP), 4D (silencing of MILIP), and 4E (overexpression of MILIP). In addition, at the left-bottom panel of Figure 3C, "MCF-7 MILIP.1" should be "MCF-7 shMILIP.1"?

16. In Figures 3D and 3E, after p53 RIP, did the authors try to explore the binding of p53 and MILIP with negative control RNA, like GAPDH/U1/U6, and positive control RNA binding to p53 protein?

17. In Figure 4H, why the authors performed the experiment under DOX treatment, but not in Figure 4F?

Minor concerns:

1. A positive control using previous identified Myc regulated lncRNAs may need to be include in figure 1b, 1c.

2. From figure 1b and 1g, after knockdown of Myc, fold change of downregulation of MILIP are not consistent. Figure 1b showed three times downregulated, but not in figure 1g.

3. Extended figure 3d, the label of BBC 3 should change to PUMA?

4. Figure 2d, authors showed an overall survival of lung cancer and bladder cancer, breast or colon cancer need to be included since they are the major cancer types authors studied.

5. Does MILIP affect TRIMPL2 expression?

Reviewer #2 (Remarks to the Author):

This study investigated a potential role of a lncRNA MILIP in driving cancer pathogenesis via regulation of MYC and p53. The authors convincingly show that this lncRNA also called MAFG-AS1, is strongly upregulated in lung and colon adenocarcinoma and this upregulation is mediated by MYC. Results from mouse experiments and patient survival analysis further indicate the oncogenic potential of this lncRNA and the association with poor prognosis in some cancers. However, the data on the downstream mechanism by which this lncRNA acts in cells is not convincing and as outlined below, the direct regulation of p53 by this lncRNA is too weak to claim that p53 directly binds to MILIP and regulates p53 degradation. Due to these reasons and a number of technical concerns, interest in this study is largely diminished.

Major Concerns

1. p53 protein is expressed at expressed in the range of 21,000 to 160,000 molecules per cell depending on the cell line

(<https://bionumbers.hms.harvard.edu/bionumber.aspx?id=106910&ver=7>). In the context of DNA damage, these levels would increase 2-10 fold. For the lncRNA MILIP to bind to p53 and destabilize p53, it needs to be expressed at stoichiometric levels. I do not believe that there is any lncRNA expressed in any cell at levels even 10-fold less than 21,000. Thus, the evidence presented here is strong enough to claim that p53 directly binds to MILIP and regulates p53 degradation. It is more than likely that the effects on p53 levels, ubiquitination, etc observed by the authors, are an experimental artifact.

2. lncRNA function is linked to their subcellular localization. The authors never examined the subcellular localization of MILIP by subcellular fractionation or RNA-FISH. This is a huge concern as lncRNA localization is linked to function and can limit the pathways/proteins it is able to interact with. Does MILIP co-localize with p53?

3. Fig. 3G: Experiments involving very strong overexpression of MILIP such as in this figure are unlikely to accurately represent phenomena that occur at the basal level. Was Flag-p53 used here because strong MILIP overexpression resulted in loss of endogenous p53? Also, the authors never checked if the overexpressed MILIP using pcDNA3 localizes to the correct subcellular compartment. This is an important point because it is known that nuclear lncRNAs often localize to the cytoplasm in overexpression experiments (PMID:25378317).

4. The authors need to initially show the effect of MILIP overexpression/knockdown on p53 levels independent of c-Myc knockdown/overexpression.

5. The method of MILIP knockdown (by siRNAs or shRNAs) is inconsistent throughout the paper. In some figures only one siRNA or shRNA is used without a rationale why one siRNA is used over the other.

6. In Fig. 2C, should p53 knockdown result in increased clonogenicity compared to the control? Also, only one p53 siRNA is used and knockdown needs to be validated by western blot. Additionally, in Fig. 2E-H, the authors should show the effect of MILIP knockdown alone on anchorage-independent growth.

7. The authors should check if MILIP expression is in turn regulated by p53 or p53-related pathways, either at the transcriptional or post-transcriptional level.

8. The authors chose to examine the post-translational regulation of p53 by TRIML2 because TRIML2 was the most upregulated p53 target following MILIP knockdown. The rationale for choosing this protein could have been stronger. It would have been preferable if the authors had performed p53 Co-IPs followed by mass spectrometry with or without MILIP knockdown to identify p53 interactors whose interaction with p53 is affected by MILIP expression. Additionally, it is not clear whether the effect of MILIP knockdown on TRIML2 is exclusively due to decreased p53 levels or if there is an additional regulatory mechanism involved.

9. The authors need to validate the effect of MILIP knockdown on TRIML2 observed in the RNA-seq data by qPCR and western blotting. Fig 4a shows no effect of MILIP knockdown on TRIML2.

10. Do c-Myc levels affect p53 SUMOylation/ubiquitination/stability? This would be consistent with the story that c-Myc negative regulation of p53 is through MILIP regulation of p53.

11. To further prove that c-Myc transcriptionally regulates MILIP, the authors could mutate the endogenous c-Myc binding region in the MILIP promoter and then measure the effect on MILIP expression.

Minor Concerns

1. The authors could report the evolutionary conservation of MILIP as conservation is linked to functionality. It would also indicate the potential to study MILIP in animal models.

2. Fig. 3A: (1) The difference in p53 half-life is not very striking due to the differences in band densities. Additionally, the authors should specify if this difference in band density is the result of MILIP knockdown affecting p53 levels. (2) Authors should report statistics from the three experiments in the quantification to show reproducibility.

3. Fig. 4i: Authors should show statistics from three experiments.

4. Line 89 and 153: "mostly" should be "most"

5. Figure 3C: "MCF-7 MILIP.1" should be "MCF-7 shMILIP.1"

6. It may be helpful to specify that the MILIP shRNAs are DOX-inducible for clarity.

7. Lines 34, 36, and 37 – should papers be cited in the abstract?
8. Line 46 – switch colon to a period
9. Fig. 1e – define RS (RNA-seq?)
10. Line 89 and 220– mostly to most
11. Lines 97-99 – This sentence is worded in a confusing way. Instead – p53 expression was increased upon MILIP knockdown and decreased upon overexpression.

Reviewer #3 (Remarks to the Author):

In this manuscript, Feng et al. present data characterizing the biological function and regulation of a c-Myc-responsive, pan-cancer-associated lncRNA that they name MILIP. By using combined bioinformatics analysis, cellular and molecular biology and biochemistry approaches, the authors demonstrated that 1), MILIP is increased in expression in diverse types of cancers and its expression is transcriptionally driven by the proto-oncogene c-Myc; 2) MILIP represses the expression of the tumour suppressor p53 and promotes cancer cell viability and tumorigenicity; 3), MILIP is physically associated with p53 and promotes p53 polyubiquitination and degradation through inhibiting p53 modification by SUMO-2/3.

Cancer cells that harbour wild type p53 often express low levels of the protein. Although this is commonly accounted for by the loss of the tumour suppressor ARF that leads to increased ubiquitination of p53 by MDM2, the results presented by these authors propose the existence of an alternative lncRNA mechanism that directly links c-Myc to repressing p53. The results would be of broad interest to those in the fields of cell biology and cancer biology. Overall, this study is original, and data shown are in general of good quality. Addressing a few concerns listed below would further strengthen the conclusions.

Major points:

1. As p53 proteins are localized in both nucleus and cytoplasm, the authors should investigate the intracellular localization of the lncRNA MILIP as well.
2. What are the copy numbers of MILIP RNA molecular in normal and cancer cell lines?
3. Does MILIP regulate its neighbour gene MAFK expression in A549 and MCF-7 cells?
4. Does MILIP have homologs in other species?
5. Deletion mapping experiments identified that the -966/-1199 segment within exon 2 of MILIP is required for its binding to p53. It is not clear why the authors select this segment? Any particular nucleotide sequence(s) or structure(s)?
6. Please provide detailed methods for CRISPR interference (CRISPRi) as the current description is too brief.

Minor points:

1. Figure 4g could be moved to supplementary figures.
2. In Extended Data Figure 4d, the backgrounds of figures are not consistent. For instance, the background of A549 shMILIP.2 cells under DOX treatment is too bright in comparison with same cells under UT and DOX withdraw conditions.

Response to Reviewer #1

We thank this reviewer's constructive comments.

1. The main finding of this study is lncRNA MILIP. However, the sequence of MILIP and isoforms of MILIP are not reported. qRT-PCR analysis of MILIP expression and copy number in all the cell lines used should be listed. More structure information related to MILIP needs to be shown.

As suggested, we have provided MILIP sequence and isoform information in Supplementary Fig. 4h, i. Accordingly, sentences have been added in text to describe this (lines 118-121, page 5).

We have conducted qRT-PCR analysis of MILIP expression and copy numbers in all cell lines used in this study. The results are now shown in Supplementary Fig. 3a, b. Accordingly, sentences have been added in text to describe this (lines 99-104, pages 4-5).

We have also shown more MILIP structure information in Supplementary Fig. 4j.

2. The localization of MILIP is very important, but it is not shown. From the extended figure 10a, authors seem to imply MILIP as a cytoplasm lncRNA. The localization needs to be clarified, and the location of sumoylation also needs to be identified.

Analysis of subcellular fractions by qPCR revealed that ~60-70% MILIP was cytoplasmic, whereas small amounts of MILIP located to the nucleus in A549 and MCF-7 cells (Supplementary Fig. 11a, b). Similarly, ISH analysis of cells grown in coverslips also showed cytoplasmic and nuclear localization of MILIP (Supplementary Fig. 3e). Investigation of the location of p53 SUMOylation found that SUMO-2/3-modified p53 was readily detected in the cytoplasmic but not nuclear fractions (Supplementary Fig. 13g). Sentences have been added in text to describe these (lines 207-210, page 8; lines 266-269, page 10).

3. At the beginning of the analysis, the rationale of choosing to study of MILIP is not clear.

We have rewritten the first paragraph of main text to provide a clearer rationale of "choosing to study of MILIP". The paragraph is now read as "Since aberrant c-Myc activation drives key gene networks contributing to the

pathogenesis of many human cancers, and an increasing number of lncRNAs have been found to be involved in c-Myc-mediated signaling, we sought to identify lncRNA effectors downstream of c-Myc that may play roles in cancer development and progression in the pan-cancer context. Through interrogating the lncRNA expression data in the TCGA, we identified a panel of lncRNAs that were upregulated in at least 18 of 20 cancer types in relation to corresponding normal tissues (Supplementary Fig. 1a). Among the five most prominently upregulated lncRNAs were two previously reported c-Myc-responsive lncRNAs, CONCR (DDX11-AS1), GAS5. A third lncRNA that we now call MILIP [also known as v-maf avian musculoaponeurotic fibrosarcoma oncogene homolog G (MAFG) antisense RNA1 (MAFG-AS1)] also contains a consensus c-Myc-binding region (c-Myc-BR) in its gene promoter (Supplementary Fig. 1a, b, 2a). We therefore set to investigate whether MILIP is indeed regulated by c-Myc and whether it is involved in cancer pathogenesis” (lines 64-75, pages 3-4).

4. Figure 1a, ChIP-PCR result is not enough; ChIP-seq should be performed and show the Myc binding region at the promoter.

Analysis of chromatin immunoprecipitation sequencing (ChIP-seq) data from the Encyclopedia of DNA Elements (ENCODE) Consortium revealed c-Myc and Max, which is necessary for c-Myc-mediated transcriptional activation (Amati B, *et al.*, *Cell*. 1993; 72: 233-245), binding peaks at the c-Myc-BR of the MILIP promoter (Supplementary Fig. 2a). A sentence has been added in text to describe this (lines 77-80, page 4). We conducted ChIP-PCR to further confirm that c-Myc indeed binds to the c-Myc-BR (Fig. 1a, Supplementary Fig. 2b). We trust that in conjunction with a number of other assays as shown in Fig. 1b, c, Supplementary Fig. 2c-j, the data are sufficient to demonstrate that c-Myc transcriptionally regulates MILIP expression.

5. In vitro study for the cell line choices are not clear. The authors identified MILIP as a pan-cancer lncRNA, but Figure 1 mostly focuses on lung and colon cancers. Later in figure 1h and figure 2, the authors move to breast cancer. Colon cancer cell line also need to be included in the later studies.

We appreciate this comment and the potential for distracting the reader. MILIP was upregulated in 18/20 cancer types (including lung, breast and colon; Supplementary Fig. 1) and all major experiments presented throughout the study were performed in the lung cancer cell line A549 and the breast cancer cell line MCF-7. We now present data from A549 and MCF-7 in all figures to ensure cell line consistency for the functional and biological work. Other data concerning the colon cancer cell line HCT116 has been transferred to the Supplementary Fig. 2b, c, g, Supplementary Fig. 3c, d. Note that HCT116 colon cancer cell data remains as part of the initial discovery process but it is now made clear from the beginning that A549 and MCF-7 cells are being used as models of p53 function throughout the study.

6. Previous studies showed that TRIML2 induced p53 is associated with apoptosis. The authors checked the caspase proteins but did not perform any apoptosis-related experiment.

We have quantitated apoptosis induced by knockdown of MILIP using propidium iodide and annexin V staining. The results showed that knockdown of MILIP indeed caused apoptosis in A549 and MCF-7 cells, consistent with our results showing caspase-3 activation and PARP cleavage. These results are now shown in Supplementary Fig. 7a, b.

7. In figure 2, most of the data showed by knockdown of MILIP, as an oncogene, overexpression data is also required to address the phenotype of this lncRNA. Similarly, the mouse model only showed knockdown MILIP xenograft, overexpressing mouse model is missing.

Fig. 2h and Supplementary Fig. 8h do show that overexpression of MILIP in MCF10A mammary epithelial cells, which express relatively low levels MILIP, increased their anchorage-independent growth. For completeness, we also tested the effects of MILIP overexpression on A549 and MCF-7 cell growth. The results showed that

overexpression of MILIP indeed increased, albeit moderately, their clonogenic efficiency (Supplementary Fig. 6g). Accordingly, sentences have been added/modified in text to indicate this (lines 172-175, page 7; line 152, page 6).

We also carried out additional mouse experiments, in which we discontinued treatment with doxycycline (DOX) in one group of mice to restore the expression of MILIP caused by induced knockdown. We found that cessation of DOX treatment led to recovery of MILIP expression and re-growth of the xenografts of A549 cells carrying inducible MILIP shRNA in response to DOX. These results are now shown in Fig 2b, Supplementary Fig. 6e, f. Accordingly, sentences have been added in text to indicate this (lines 153-154, page 6).

8. Extended figure 5a, authors showed overall survival probability in KIRC from TCGA, but this cancer type does not have upregulated MILIP expression shown in extended figure 1a, which lead to the confusion to this reviewer. Also, ACC is not included in extended figure 1a.

We apologise for this oversight and have removed survival probability data of KIRC and ACC from extended figure 5a (now Supplementary Fig. 8a) and substituted survival probabilities with data from TCGA-BLCA showing similar findings.

9. Figure 3c, the input expression of MDM2 in A549 cell lines are very different. The shMILIP.1 part showed a very high input expression of MDM2, but in the shMILIP.2 part, there is basically no input expression of MDM2. Also, the IgG sample in two trials are also different.

We have re-conducted experiments shown in Fig. 3c and new figures with improved quality that show comparable levels of MDM2 in the input samples are now presented. These figures also consistently show that no MDM2 was precipitated in the IgG control samples.

10. From the manuscript, knockdown MILIP decreased expression of p53 while upregulation of TRIML2 stabilize and induce p53 expression level. However, this phenomenon was not shown in several IP experiments include figure 4e IP and input p53 expression, 4h input p53 expression level, and extended figure 9e input and IP p53 expression.

Fig. 4e shows that overexpression of TRIML2 reduces p53 polyubiquitination. Figure 4h (now Fig. 4g) shows that MILIP silencing increases the amount of TRIML2 co-immunoprecipitating with p53, whereas extended figure 9e (now Supplementary Fig. 13e) shows that TRIML2 overexpression reduces the amount of MILIP associating with p53. To facilitate these experiments, we blocked proteasomal degradation of p53 by adding the proteasome inhibitor MG132 as labelled in the figures now, which resulted in similar levels of p53 in different samples. This is a common practice for carrying out similar experiments in examining posttranslational modification of p53 (Yuan J, *et al.*, *Cell*. 2010; 140: 384-396; Shi D, *et al.*, *Proc Natl Acad Sci U S A*. 2009; 106: 16275-16280; Yan S, *et al.*, *Oncogene*. 2014; 33: 5424-5433).

11. The protein stability of p53 is not significantly induced after MILIP treatment in some figures.

Data shown in Fig. 3a clearly indicated that silencing of MILIP increased the protein stability of p53. As mentioned above, we treated cells with the proteasome inhibitor MG132 to stabilize the p53 protein in experiments testing posttranslational modification of p53. In these experiments, the levels of p53 in cells with or without silencing of MILIP appeared similar due to the addition of MG132.

12. What is the expression of MILIP in A549, MCF-7, HCT116, HME-1, MCF10A, and other (cancer) cell lines? MYC plays an important role in physiological conditions. Will MYC induce MILIP expression in normal tissue? What is MILIP's role in physiological condition?

The expression of MILIP in all the cell lines used in this study is now shown in Supplementary Fig. 3a. Accordingly, sentences have been added in text to indicate this (lines 99-102, page 4).

Overexpression of c-Myc upregulated MILIP in MCF10A and HME-1 cells, indicating that the increased levels of c-Myc can drive MILIP expression in normal tissues. However, siRNA knockdown of c-Myc did not cause any significant changes in the MILIP levels, implicating that mechanisms other than c-Myc may also be involved in regulating the expression of MILIP in normal cells. These results are now shown in Supplementary Fig. 2l, m. Accordingly, sentences have been added in text to indicate this (lines 93-97, page 4).

We do not entirely understand the role of MILIP under physiological conditions. Nevertheless, silencing of MILIP did not cause any changes in the viability of MCF10A and HME-1 cells, suggesting that MILIP did not have a major role in regulating cell survival and proliferation in normal cells. These results are now shown in Supplementary Fig. 8d. Accordingly, sentences have been added in text to indicate this (lines 168-170, page 7). As shown in Fig. 2g and h, knockdown of MILIP prevented c-Myc overexpression-driven anchorage-independent growth of MCF10A cells, whereas overexpression of MILIP triggered anchorage-independent growth of MCF10A cells. These results suggest that MILIP can function to promote neoplastic transformation driven by high levels of c-Myc. This is stated in text (lines 172-176, page 7).

13. MILIP is located to the same genomic locus as the protein-coding gene, MAFG. Does c-Myc regulate the transcriptional level of MAFG and MAFG also promotes the protein stability of p53?

Neither knockdown nor overexpression of c-Myc altered the expression of MAFG mRNA in A549 and MCF-7 cells, suggesting that c-Myc does not have a role in transcriptionally regulating MAFG expression under the conditions of our experiments. These results are now shown in Supplementary Fig. 4f, g. Accordingly, sentences have been added in text to indicate this (lines 113-116, page 5).

We have also found that knockdown of MAFG did not alter the protein stability of p53 in A549 cells (Supplementary Fig. 9f), ruling out the possibility that MAFG contributes to the regulation p53 expression. Accordingly, sentences have been added in text to indicate this (lines 191-193, page 8).

14. Did the authors evaluate the effect of MILIP on neoplastic transformation with overexpression of MYC in MCF10A cells of Figure 2H? Only overexpression of MILIP promoting neoplastic transformation shows MILIP could strongly regulate cell growth.

We indeed evaluated the effect of MILIP on neoplastic transformation with overexpression of MYC in MCF10A cells. The results are shown in Fig. 2g, which demonstrated that knockdown of MILIP prevented c-Myc overexpression driven anchorage-independent growth of MCF10A cells.

15. In Figure 3C, the authors have confirmed that knockdown of MILIP could significantly increase the protein stability of p53, but there is no significant induction after knockdown of MILIP in Input lanes. Similar results are also shown in Figure 4A (silencing of MILIP), 4B (overexpression of MILIP), 4D (silencing of MILIP), and 4E (overexpression of MILIP). In addition, at the left-bottom panel of Figure 3C, "MCF-7 MILIP.1" should be "MCF-7 shMILIP.1"?

As mentioned above, we treated cells with the proteasome inhibitor MG132 to stabilize the p53 protein in experiments testing posttranslational modification of p53. In these experiments, the levels of p53 in cells with or without silencing or overexpression of MILIP appeared similar due to the addition of MG132. This is a common practice for carrying our similar experiments in examining posttranslational modification of p53 (Yuan J, *et al.*, *Cell*. 2010; 140: 384-396; Shi D, *et al.*, *Proc Natl Acad Sci U S A*. 2009; 106: 16275-16280; Yan S, *et al.*, *Oncogene*. 2014; 33: 5424-5433).

We have corrected the error pointed out by the reviewer. The left-bottom panel of Fig. 3c is now correctly labeled with “MCF-7 shMILIP.1”.

16. In Figures 3D and 3E, after p53 RIP, did the authors try to explore the binding of p53 and MILIP with negative control RNA, like GAPDH/U1/U6, and positive control RNA binding to p53 protein?

We included U6 as a negative control and MDMX mRNA that is known to bind to p53 as positive control in the RIP experiment now shown in Supplementary Fig. 10a (Tournillon AS, *et al.*, *Oncogene*. 2017; 36: 723-730). The results show that p53 did not bind to U6 but was associated with MDMX mRNA.

17. In Figure 4H, why the authors performed the experiment under DOX treatment, but not in Figure 4F?

The experiment shown in Figure 4h (now Fig. 4g) was conducted in cells carrying inducible MILIP shRNA in response to DOX. The cells were treated with DOX in order to induce knockdown of MILIP. However, Fig. 4f was to test the effect of overexpression of TRIML2 on p53 expression in cells without carrying inducible MILIP shRNA.

Minor concerns:

1. A positive control using previous identified Myc regulated lncRNAs may need to be include in figure 1b, 1c.

We have included the lncRNA OVAAL that is known to be regulated by c-Myc in Fig. 1b (Sang B, *et al.*, *Proc Natl Acad Sci U S A*. 2018; 115: E11661-E11670). This is stated in text (lines 80-83, page 4). We have also added OVAAL as a positive control in Fig. 1c, Supplementary Fig. 2g.

2. From figure 1b and 1g, after knockdown of Myc, fold change of downregulation of MILIP are not consistent. Figure 1b showed three times downregulated, but not in figure 1g.

We have repeated experiments shown in Fig. 1g. The results presented now in Fig.1g show the similar extent of downregulation of MILIP caused by knockdown of c-Myc as shown in Fig. 1b.

3. Extended figure 3d, the label of BBC3 should change to PUMA?

We have replaced the label of BBC3 in extended figure 3d (now Supplementary Fig. 5h) to PUMA.

4. Figure 2d, authors showed an overall survival of lung cancer and bladder cancer, breast or colon cancer need to be included since they are the major cancer types authors studied.

We have included breast cancer data as additional panels in Fig. 2d.

5. Does MILIP affect TRIMPL2 expression?

MILIP negatively affects TRIML2 expression as shown in Supplementary Fig. 5i-k.

Response to Reviewer #2

We thank this reviewer’s constructive comments.

Major Concerns

1. p53 protein is expressed at expressed in the range of 21,000 to 160,000 molecules per cell depending on the cell line (<https://bionumbers.hms.harvard.edu/bionumber.aspx?id=106910&ver=7>). In the context of DNA damage, these levels would increase 2-10 fold. For the lncRNA MILIP to bind to p53 and destabilize p53, it needs to be expressed at stoichiometric levels. I do not believe that there is any lncRNA expressed in any cell at levels even 10-fold less than 21,000. Thus, the evidence presented here is strong enough to claim that p53 directly binds to MILIP and regulates p53 degradation. It is more than likely that the effects on p53 levels, ubiquitination, etc observed by the authors, are an experimental artifact.

As a tumour suppressor, the abundance of p53 in fast-growing unstressed cancer cells is kept low (Bode AM, *et al.*, *Nat Rev Cancer*. 2004; 4: 793-805). However, past studies have documented wide variations in the absolute concentration of the p53 protein within a cell, conceivably due to different cell types and varying experimental approaches employed by different studies (Ma L, *et al.*, *Proc Natl Acad Sci U S A*. 2005; 102: 14266-14271; Beck M, *et al.*, *Mol Syst Biol*. 2011; 7: 549; Burgin E, *et al.*, *Analyst*. 2014; 139: 3235-3244). For example, measurement using the enzyme-linked immunosorbent assay (ELISA) method found that the basal p53 concentration in MCF7 cell is 15.84×10^4 molecules per cell (Ma L, *et al.*, *Proc Natl Acad Sci U S A*. 2005; 102: 14266-14271), whereas a quantitative proteomic study showed that p53 was present with 4220 molecules per U2OS cell (Beck M, *et al.*, *Mol Syst Biol*. 2011; 7: 549). More recently, a microfluidic microarray-based single molecule detection study reported that the majority of MCF7 cells contained fewer than 2500 copies of p53 per cell (Burgin E, *et al.*, *Analyst*. 2014; 139: 3235-3244).

We have carried out absolute quantitation of p53 using the ELISA assay. The results showed that there were 1422 and 2858 p53 molecules in A549 and MCF-7 cells, respectively (Supplementary Fig. 11e). Moreover, our semi-quantitative Western blot assays showed that the vast majority of p53 was in the nucleus, whereas ~10% and ~4% of the protein was detected in the cytoplasm of A549 and MCF-7 cells, respectively (Supplementary Fig. 11c). On the other hand, absolute quantitation using quantitative PCR (qPCR) demonstrated that there were ~79-152 MILIP molecules per cancer cell (Supplementary Fig. 3b). In particular, there were 152 ± 8 and 112 ± 11 MILIP molecules per A549 and MCF-7 cell, respectively. In addition, analysis of subcellular fractions by qPCR revealed that ~60-70% MILIP was cytoplasmic, whereas small amounts of MILIP located to the nucleus in A549 and MCF-7 cells (Supplementary Fig. 11a, b). Thus, the relative level of MILIP versus p53 was 109/142 ($\approx 1:1.3$) and 65/86 ($\approx 1:1.3$) in the cytoplasm of A549 and MCF-7 cells, respectively (Supplementary Fig. 11f). In contrast, the relative level ranged from 42/1280 ($\approx 1:30$) to 47/2772 ($\approx 1:102$) in the nucleus (Supplementary Fig. 11f).

In accordance with the differential subcellular localization of the MILIP and p53 molecules, depletion of MILIP through RNA pulldown abolished the presence of p53 in the cytoplasmic fractions of A549 and MCF-7 cells (Supplementary Fig. 11g), indicating sufficient stoichiometric amounts of MILIP to interact with p53 in the cytoplasm. Given that p53 polyubiquitination and proteasomal degradation primarily occurs in the cytoplasm (Hock AK, *et al.*, *Biochim Biophys Acta*. 2014; 1843: 137-149), these results suggest that the promoting effects of MILIP on p53 polyubiquitination and degradation is largely accounted for by the cytoplasmic interaction of MILIP with p53. Consistent with this, SUMO-2/3-modified p53 was readily detected in the cytoplasmic but not the nuclear fraction (Supplementary Fig. 13g), supporting the notion that MILIP primarily effects on cytoplasmic p53 to promote p53 polyubiquitination and degradation. Furthermore, we found that the SUMO E3 ligase TRIML2 localised predominantly to the cytoplasm (Supplementary Fig. 13f). Collectively, these results indicate that it is the cytoplasmic interaction between MILIP and p53 that causes inhibition of p53 SUMOylation leading to promoting p53 polyubiquitination and proteasomal degradation, and that the stoichiometric amounts of MILIP is sufficient to interact with p53 in the cytoplasm.

These results are now shown in Supplementary Fig. 3b, Supplementary Fig. 11a-c, e-g, Supplementary Fig. 13f, g, accordingly, sentences (paragraphs) have been added in text and figure legends to indicate and discuss the results (lines 207-231, pages 8-9; lines 266-269, page 10).

2. lncRNA function is linked to their subcellular localization. The authors never examined the subcellular localization of MILIP by subcellular fractionation or RNA-FISH. This is a huge concern as lncRNA localization is linked to function and can limit the pathways/proteins it is able to interact with. Does MILIP co-localize with p53?

Analysis of subcellular fractions by qPCR revealed that ~60-70% MILIP was cytoplasmic, whereas small amounts of MILIP located to the nucleus in A549 and MCF-7 cells (Supplementary Fig. 11a, b) and this finding was confirmed using in situ hybridization (ISH) (Supplementary Fig. 3e). Accordingly, sentences have been added to text to indicate this (lines 207-210, page 8).

Moreover, MILIP and p53 were co-precipitated (pulled down), suggesting that they co-localise with each other. This was supported by the colocalization of p53 and MILIP as shown in studies using immunofluorescent staining of p53 and fluorescence in situ hybridization (FISH) analysis of MILIP in cells grown on coverslips. These results are now shown in Fig. 3d-f, Supplementary Fig. 10a, b. Accordingly, sentences have been added in text to indicate this (lines 195-199, page 8).

3. Fig. 3G: Experiments involving very strong overexpression of MILIP such as in this figure are unlikely to accurately represent phenomena that occur at the basal level. Was Flag-p53 used here because strong MILIP overexpression resulted in loss of endogenous p53? Also, the authors never checked if the overexpressed MILIP using pcDNA3 localizes to the correct subcellular compartment. This is an important point because it is known that nuclear lncRNAs often localize to the cytoplasm in overexpression experiments (PMID:25378317).

We agree that supraphysiological expression levels of a protein (or lncRNA) may not accurately reflect its function at basal expression. Nevertheless, in this study the effect of MILIP on p53 expression was mainly demonstrated by MILIP knockdown experiments as shown in Fig. 3a, Supplementary Fig. 5b, and Supplementary Fig. 9b, whereas we used overexpression experiments as a complementary means.

Fig. 3g is about deletion mapping experiments, which intends to show that introduction of a MILIP mutant with the -966/-1199 segment deleted failed to downregulate p53 as stated in text (lines 203-205, page 8). We used Flag-p53 simply in order to facilitate the experiment. Nevertheless, we have also conducted the same experiment in cells without introduction of exogenous p53. The results similarly showed that introduction of the MILIP mutant with the -966/-1199 segment did not cause downregulation of p53 as did overexpression of wild-type MILIP. These results are now also shown in Fig. 3g.

Moreover, the subcellular localization of transfected MILIP (pcDNA3.1-MILIP) was found to be the same as endogenous MILIP (i.e. mostly cytoplasmic with some nuclear expression) now presented as Supplementary Fig. 11b.

4. The authors need to initially show the effect of MILIP overexpression/knockdown on p53 levels independent of c-Myc knockdown/overexpression.

As suggested, we have re-ordered text to initially show the effect of MILIP overexpression/knockdown on p53 levels is independent of c-Myc knockdown/overexpression (lines 127-145, pages 5-6). Accordingly, relevant figures have also been re-ordered.

5. The method of MILIP knockdown (by siRNAs or shRNAs) is inconsistent throughout the paper. In some figures only one siRNA or shRNA is used without a rationale why one siRNA is used over the other.

We generally used two independent siRNAs (or shRNAs) for all knockdown experiments but some figures were simplified to show one RNAi result. The MILIP siRNAs/shRNAs used throughout have similar knockdown efficiencies (Supplementary Fig. 5a, d) and there is no special rationale or bias. For completeness, as requested,

we have modified figure panels to illustrate the effects of two independent siRNAs/shRNAs throughout the study (Fig. 1h, Fig. 2c, Fig. 4a, c, d, Supplementary Fig. 5c, j, Supplementary Fig. 7a-d, Supplementary Fig. 12b-e).

6. In Fig. 2C, should p53 knockdown result in increased clonogenicity compared to the control? Also, only one p53 siRNA is used and knockdown needs to be validated by western blot. Additionally, in Fig. 2E-H, the authors should show the effect of MILIP knockdown alone on anchorage-independent growth.

We agree with the general premise that stabilizing the levels of p53, e.g. with nutlin-3, would decrease clonogenicity whereas decreasing p53 would tend to increase clonogenicity. We confirmed that siRNAs against p53 were functional (Fig. 2c, Supplementary Fig. 7d) but we did not observe significantly increased colonies in our experiments. Nevertheless, inhibition of endogenous p53 expression does not always promote cell proliferation in cancer cells (Madan E, *et al.*, *J Biol Chem.* 2018; 293: 4262-4276; Yang D, *et al.*, *Theranostics.* 2018; 8: 3517-3529) and we assume that this relates to the cell context. These experiments were not performed under stress conditions and manipulating basal (low level) p53 in cancer cells did not augment cell proliferation.

New data have been added to Fig. 2c regarding additional siRNAs targeting MILIP along with a second independent siRNA targeting p53 in Supplementary Fig. 7d. Moreover, we present control data to demonstrate the efficiency of MILIP and p53 knockdown for these experiments (Fig. 2c, Supplementary Fig. 7d).

The effects of MILIP knockdown alone on anchorage-independent growth are now shown in Supplementary Fig. 8f, g.

7. The authors should check if MILIP expression is in turn regulated by p53 or p53-related pathways, either at the transcriptional or post-transcriptional level.

We have examined whether MILIP expression is in turn regulated by p53. siRNA knockdown of p53 did not significantly impinge on the levels of MILIP, suggesting that p53 does not play a role in regulating MILIP expression. These data are now shown in Supplementary Fig. 5m. Accordingly, sentences have been added in text to indicate this (lines 136-138, page 6).

8. The authors chose to examine the post-translational regulation of p53 by TRIML2 because TRIML2 was the most upregulated p53 target following MILIP knockdown. The rationale for choosing this protein could have been stronger. It would have been preferable if the authors had performed p53 Co-IPs followed by mass spectrometry with or without MILIP knockdown to identify p53 interactors whose interaction with p53 is affected by MILIP expression. Additionally, it is not clear whether the effect of MILIP knockdown on TRIML2 is exclusively due to decreased p53 levels or if there is an additional regulatory mechanism involved.

To further investigate the mechanism of MILIP-mediated polyubiquitination and degradation of p53. Intriguingly, three of the top 10 most upregulated genes following MILIP knockdown were TRIML2, NUPR1 and HDAC5, all known to interact with p53 and regulate p53 activity (Supplementary Fig. 12a) (Kung CP, *et al.*, *Mol Cancer Res.* 2015; 13: 250-262; Clark DW, *et al.*, *Curr Cancer Drug Targets.* 2008; 8: 421-430; Sen N, *et al.*, *Mol Cell.* 2013; 52: 406-420). However, silencing of TRIML2 but not NUPR1 and HDAC5 reduced the expression of p53 in A549 cells (Supplementary Fig. 12b), proposing that TRIML2 may be involved in the regulation of p53 by MILIP. Indeed, TRIML2 and p53 were reciprocally coprecipitated, and more importantly, the association between TRIML2 and p53 was markedly increased in cells with MILIP knocked down. These results are now shown in Supplementary Fig. 12a, b, Fig. 4g, Supplementary Fig. 13a, b. Accordingly, we have rewritten sentences/paragraphs in text to better justify the rationale for examination of the post-translational regulation of p53 by TRIML2 (lines 235-240, page 9).

We have also examined the role of p53 in upregulation of TRIML2 caused by MILIP knockdown. The results showed that co-knockdown of p53 diminished the upregulation of TRIML2 caused by MILIP knockdown, indicating that upregulation of TRIML2 in cells with MILIP knockdown is primarily due to a p53-mediated increase. These results are now shown in Supplementary Fig. 13h. Accordingly, sentences have been added in text to indicate this (lines 269-271, page 10).

9. The authors need to validate the effect of MILIP knockdown on TRIML2 observed in the RNA-seq data by qPCR and western blotting. Fig 4a shows no effect of MILIP knockdown on TRIML2.

As requested, we have validated the effect of MILIP knockdown on TRIML2 expression using qPCR and Western blot analysis (Supplementary Fig. 5i, j). Moreover, we have replaced Fig. 4a with a panel that better reflects the upregulation of TRIML2 following MILIP knockdown. Accordingly, sentences have been added in text to indicate this (lines 132-135, pages 5-6).

10. Do c-Myc levels affect p53 SUMOylation/ubiquitination/stability? This would be consistent with the story that c-Myc negative regulation of p53 is through MILIP regulation of p53.

We have examined the effect of c-Myc silencing on p53 SUMOylation/ubiquitination. The results showed that silencing of c-Myc caused the increase in p53 SUMOylation, the decrease in p53 polyubiquitination and the increase in p53 stability, thus phenocopying the effects of MILIP silencing. These results are now shown in Fig. 5a-f. Accordingly, sentences/paragraphs have been added in text to indicate this (lines 274-280, pages 10-11).

11. To further prove that c-Myc transcriptionally regulates MILIP, the authors could mutate the endogenous c-Myc binding region in the MILIP promoter and then measure the effect on MILIP expression.

As suggested, we have carried out CRISPR/Cas9-mediated deletion of the c-Myc-BR in the promoter of endogenous *MILIP* gene in A549 cells. The results showed that deletion of the c-Myc-BR at the endogenous *MILIP* gene promoter diminished the expression of MILIP, which could not be rescued by c-Myc overexpression, thereby consolidating the role of c-Myc in the transcriptional activation of MILIP. These results are now shown in Supplementary Fig. 2i, j. Accordingly, sentences have been added in text to indicate this (lines 88-91, page 4).

Minor Concerns

1. The authors could report the evolutionary conservation of MILIP as conservation is linked to functionality. It would also indicate the potential to study MILIP in animal models.

We include information about the evolutionary conservation in text. It is stated that “Through bioinformatics analysis, we identified a transcript in Pan troglodytes that is highly homologous to human MILIP with 93% sequence similarity, suggesting evolutionary conservation of MILIP between primate species. However, the lack of similarity between human MILIP and *Mus musculus* transcripts makes it infeasible to test the role of MILIP in cancer initiation in transgenic mouse models” (lines 295-299, page 11). These findings are also illustrated in Supplementary Table 3.

2. Fig. 3A: (1) The difference in p53 half-life is not very striking due to the differences in band densities. Additionally, the authors should specify if this difference in band density is the result of MILIP knockdown affecting p53 levels. (2) Authors should report statistics from the three experiments in the quantification to show reproducibility.

We have repeated the experiments shown in Fig. 3a. Better panels are now shown in this figure that clearly demonstrate the difference in p53 half-life between cells with or without MILIP knockdown. Statistics from the three experiments in the quantification is also clearly shown to demonstrate reproducibility.

3. Fig. 4i: Authors should show statistics from three experiments.

As suggested, statistics from three experiments is shown in Fig. 4i (now Fig. 4h).

4. Line 89 and 153: “mostly” should be “most”

We have corrected this error.

5. Figure 3C: “MCF-7 MILIP.1” should be “MCF-7 shMILIP.1”

We have corrected this error.

6. It may be helpful to specify that the MILIP shRNAs are DOX-inducible for clarity.

As suggested, we have clarify that the MILIP shRNAs are DOX-inducible in text (lines 128-132, page 5).

7. Lines 34, 36, and 37 – should papers be cited in the abstract?

We have removed the citations from the abstract.

8. Line 46 – switch colon to a period

We have changed the colon to a period.

9. Fig. 1e – define RS (RNA-seq?)

We have defined RS as reactive score in the Materials and Methods as well as in legends to this figure.

10. Line 89 and 220– mostly to most

We have changed “mostly” to “most” in these lines (line 128, page 5; line 666, page 26)

11. Lines 97-99 – This sentence is worded in a confusing way. Instead – p53 expression was increased upon MILIP knockdown and decreased upon overexpression.

This sentence has been removed.

Response to Reviewer #3

We thank this reviewer’s constructive comments.

Major points:

1. As p53 proteins are localized in both nucleus and cytoplasm, the authors should investigate the intracellular localization of the lncRNA MILIP as well.

Analysis of subcellular fractions by qPCR revealed that ~60-70% MILIP was cytoplasmic, whereas small amounts of MILIP located to the nucleus in A549 and MCF-7 cells (Supplementary Fig. 11a, b) and this finding was confirmed using in situ hybridization (ISH) (Supplementary Fig. 3e). Accordingly, sentences have been added to text to indicate this (lines 207-210, page 8).

2. What are the copy numbers of MILIP RNA molecular in normal and cancer cell lines?

Absolute quantitation by qPCR demonstrated that there were ~79-152 MILIP molecules per cancer cell compared with ~23-30 MILIP molecules per normal mammary epithelial cell. In particular, there were 152 ± 8 and 112 ± 11 MILIP molecules per A549 and MCF-7 cell, respectively. This result is shown in Supplementary Fig. 3b.

3. Does MILIP regulate its neighbour gene MAFG expression in A549 and MCF-7 cells?

MILIP does not regulate its neighbour gene MAFG expression, as neither knockdown nor overexpression of MILIP impinged upon MAFG expression in A549 and MCF-7 cells. These results are now shown in Supplementary Fig. 4b, c. Accordingly, sentences have been added to text to indicate this (lines 109-111, page 5).

4. Does MILIP have homologs in other species?

Through bioinformatics analysis, we identified a transcript in Pan troglodytes that is highly homologous to human MILIP with 93% sequence similarity (Supplementary Table 3), suggesting evolutionary conservation of MILIP between primate species. However, the lack of similarity between human MILIP and Mus musculus transcripts makes it infeasible to test the role of MILIP in cancer initiation in transgenic mouse models. This information is now presented in Supplementary Table 3 and described in text (lines 295-299, page 11).

5. Deletion mapping experiments identified that the -966/-1199 segment within exon 2 of MILIP is required for its binding to p53. It is not clear why the authors select this segment? Any particular nucleotide sequence(s) or structure(s)?

After we found that MILIP bound to p53 through its exon 2 (E2), we divided E2 into three randomly defined fragments. Further deletion mapping showed that the -966/-1199 segment was critical for the binding of MILIP to p53. We have added sentences in text to clarify this (lines 199-202, page 8).

6. Please provide detailed methods for CRISPR interference (CRISPRi) as the current description is too brief.

Detailed methods for CRISPR interference (CRISPRi) are now described in Material and Methods (lines 361-367, pages 13-14).

Minor points:

1. Figure 4g could be moved to supplementary figures.

As suggested, we have moved Figure 4g to Supplementary Fig. 12g.

2. In Extended Data Figure 4d, the backgrounds of figures are not consistent. For instance, the background of A549 shMILIP.2 cells under DOX treatment is too bright in comparison with same cells under UT and DOX withdraw conditions.

More visually consistent results have been substituted in the figure as requested (now Supplementary Fig. 6d).

REVIEWER COMMENTS

Reviewer #1 (Remarks to the Author):

The authors have addressed majority of my previous concerns. Several remaining/new concerns need to be addressed are listed below.

1. Colocalization of MILIP and p53 protein in Supplementary Figure 10b is less convincing: DAPI staining for nucleus is diffusing and the staining size of MILIP-p53 (yellow) is inconsistent with MILIP staining only (green) (different magnificence or exposure time?)
2. The authors concluded that "This regulation appears independent of ARF as the cell lines used (A549, MCF-7 and HCT116) were deficient in ARF expression." in Page 6 Lines 146-148. However, there is few results to show ARF deficiency is required for c-Myc to inactivate p53 through MILIP. In addition, what is the status of ARF in MDA-MB-231 and U2OS cell lines?
3. In Supplementary Figures 5c and 5f, the titles of x-axis seems not to be correct.

Reviewer #2 (Remarks to the Author):

The authors have addressed my concerns.

Reviewer #3 (Remarks to the Author):

The revised manuscript provides strong evidences of lncRNA MILIP involved in the process of c-Myc inactivation of p53. Authors have satisfactorily answered all my queries and concerns, and I recommend it to be accepted for publication in NC.

NCOMMS-19-39408A: c-Myc inactivation of p53 through the pan-cancer lncRNA MILIP drives cancer pathogenesis

Response to comments provided by Reviewer #1

We thank this reviewer's constructive comments.

1. Colocalization of MILIP and p53 protein in Supplementary Figure 10b is less convincing: DAPI staining for nucleus is diffusing and the staining size of MILIP-p53 (yellow) is inconsistent with MILIP staining only (green) (different magnification or exposure time?)

We apologize for the confusion this figure had caused. We have re-prepared this figure using previously captured microscopic images. The figure now shows better DAPI staining for the nucleus and the staining of MILIP in the overlaid panel (MILIP-p53, yellow) is consistent with MILIP staining only as indicated in the figure.

2. The authors concluded that "This regulation appears independent of ARF as the cell lines used (A549, MCF-7 and HCT116) were deficient in ARF expression." in Page 6 Lines 146-148. However, there is few results to show ARF deficiency is required for c-Myc to inactivate p53 through MILIP. In addition, what is the status of ARF in MDA-MB-231 and U2OS cell lines?

We do not intend to claim that "ARF deficiency is required for c-Myc to inactivate p53 through MILIP". Instead, our results demonstrate that, as indicated by the reviewer, "this regulation (c-Myc repression of p53) appears independent of ARF as the cell lines used (A549, MCF-7 and HCT116) were deficient in ARF expression". Both MDA-MB-231 and U2OS lines are also deficient in ARF expression (Green JL, *et al.*, *Mol Cancer Ther.* 2019; 18: 771-779; Kwong RA, *et al.*, *Clin Cancer Res.* 2005; 11: 4107-4116; Miller PJ, *et al.*, *Hum Mutat.* 2011; 32: 900-911; Yuan XW, *et al.*, *Cancer Biol Ther.* 2007; 6: 1074-1080). Phrases and references have been added in text to indicate this (lines 145-146, page 6 and lines 565-572, page 20).

3. In Supplementary Figures 5c and 5f, the title of x-axis seems not to be correct.

Supplementary Figures 5c and 5f were carried out using PG13-luc-p53-BS plasmids containing consensus p53-binding sequences and corresponding control empty plasmids PGL3 (Addgene). We have modified labelling of x-axes of these figures to clearly indicate this. Accordingly, we have modified legends to these figures.